# Personalized Federated Learning of Probabilistic Models: A PAC-Bayesian Approach

## Abstract

Federated learning aims to infer a shared model from private and decentralized data stored locally by multiple clients. Personalized federated learning (PFL) goes one step further by adapting the global model to each client's data, enhancing the model's fit for different clients. A significant level of personalization is required for highly heterogeneous clients, but can be challenging to achieve especially when they have small datasets. To address this problem, we propose a PFL algorithm named *PAC-PFL* for learning probabilistic models within a PAC-Bayesian framework that utilizes differential privacy to handle data-dependent priors. Our algorithm collaboratively learns a shared hyper-posterior and regards each client's posterior inference as the personalization step. By establishing and minimizing a generalization bound on the average true risk of clients, PAC-PFL effectively combats overfitting. Empirically, PAC-PFL achieves accurate and well-calibrated predictions as demonstrated through experiments on a highly heterogeneous dataset of photovoltaic panel power generation and the FEMNIST dataset (Caldas et al., 2019).

## 1 Introduction

*Federated Learning (FL)* aims at collaboratively learning from multiple private datasets stored by end-devices, called *clients* (Konečný et al., 2015). The main line of FL iteratively trains a *global* model that performs well for all clients. A trusted *server* orchestrates the training by sending the model to a subset of the clients at each iteration and collecting model updates from them. The server prepares the model for the next iteration by aggregating these local updates. The server only accesses the communicated model updates and not the clients' datasets. Thus, FL offers better data privacy and lowers communication than a data-centric approach, where all clients' datasets are sent to the server.

A defining characteristic of FL is that the clients' data are potentially heterogeneous (Kairouz et al., 2021; Li et al., 2020a). This violates the critical i.i.d samples assumption in training a single global model, leading to convergence issues or poor performance (Li et al., 2020b). *Personalized FL* (PFL) addresses this challenge by incorporating a further *personalization* step of adapting the global model to each client's data. While PFL has been extensively studied, the following challenges ($c1$-$c4$) remain less explored. First, ($c1$) the majority of PFL approaches yield point estimates of the models, limiting their ability to quantify epistemic uncertainty, which is problematic in safety-critical applications (Guo et al., 2017; Achituve et al., 2021). Second, ($c2$) the resulting personalized models remain closely tied and are not fully capable of capturing highly heterogeneous, multimodal scenarios. Third, ($c3$) many proposed methods suffer from performance degradation when clients' datasets are small. Finally, ($c4$) most approaches do not take into account the collection of new data over time. In this paper, we propose a PFL algorithm, called *PAC-PFL*, for tackling the challenges outlined above, ($c1$-$c4$).

A natural PFL method for probabilistic models is to collaboratively learn a shared prior and regard the posterior inference as the personalization step. However, in this setting, the learned prior depends on each client's data, conflicting with the Bayesian framework (Box and Tiao, 1992). To overcome this, we leverage a PAC-Bayesian technique capable of handling data-dependent priors (Rivasplata et al., 2020) for inferring the posterior and providing uncertainty quantification, hence addressing ($c1$). However, the clients' posterior distributions tend to resemble each other due to their connection with the shared prior, impairing the performance in heterogeneous scenarios (see $c2$). As an alternative, we collaboratively learn a shared distribution over priors, termed a *hyper-posterior*, from which the clients sample their priors to perform personalized posterior inference. By removing the linkage of

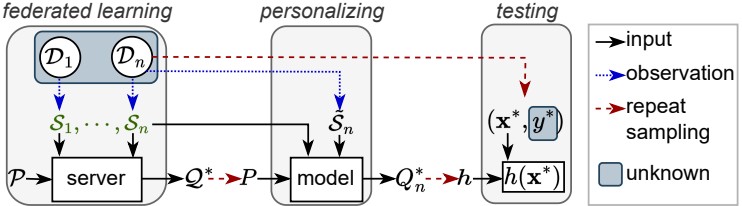

Figure 1: Overview of the framework for client $n$ using the notation introduced in Section 4. Black arrows indicate algorithmic dependencies, blue arrows show sampling from distributions, red arrows represent repeated sampling for computing expectations, and dark-shaded regions contain unknown elements. As can be seen in the figure, the prior depends on the client's data, $\mathcal{S}_n$, making it data-dependent. Note that the server does not have direct access to $\mathcal{S}_1, \cdots, \mathcal{S}_n$, but rather to some gradients, as will be discussed in Section 5. The information exchange between the clients and the server is described by Algorithm 1, which complements the conceptual dependencies illustrated in this figure.

the personalized posteriors to the same shared prior, this approach enhances flexibility in capturing heterogeneous patterns, thus resolving ($c2$).

To mitigate potential overfitting in low-data scenarios (see $c3$), we select the hyper-posterior by minimizing a generalization bound on clients' true loss. Section 4 will demonstrate that this approach provides a principled regularization of the hyper-posterior, facilitating the learning of complex models without overfitting. Regarding ($c4$), PAC-PFL accommodates collecting new data over time. Due to practical constraints, only a fraction of clients communicate with the server in each iteration. Before reconnecting with the server, a client may collect new data that can be used for personalization, but not for training the shared hyper-posterior. In an extreme case, there may be clients joining the system later, denoted as *new clients*, who have never communicated with the server. The distinction between existing and new clients is important, as distinct generalization bounds hold for each group.

We evaluate our framework on Gaussian Process (GP) regression and Bayesian Neural Network (BNN) classification as representative examples of probabilistic models. Our experiments demonstrate that PAC-PFL produces accurate and well-calibrated predictions ($c1$) even in highly heterogeneous ($c2$) and data-poor ($c3$) scenarios. Additionally, we show that PAC-PFL enables positive transfer learning from existing clients to new clients. ($c4$) Finally, our method can be interpreted through Jaynes' principle of maximum entropy (Jaynes, 1957) as explained in Appendix 8.1.

## 2 RELATED WORK

**Meta-PFL.** PAC-PFL falls under the meta-PFL category (Kulkarni et al., 2020). Meta-learning, developed independently of FL, involves training a global model on various related learning problems (tasks), which can be efficiently fine-tuned for a task. The link between meta-learning and PFL was first explored in Jiang et al. (2019), where a method was proposed to simultaneously learn the global model and its personalization. Several methods aim to find a model initialization that performs well after personalization by individual clients using their respective data (Khodak et al., 2019; Fallah et al., 2020; Chen et al., 2018). However, the approach of Khodak et al. (2019) suffers from scalability issues due to its reliance on sequential updates from clients. On the other hand, Fallah et al. (2020) proposes an intuitive algorithm by narrowing the personalization step down to taking one or a few steps of gradient descent with respect to clients' own data. However, the personalized models have a high resemblance to the shared model due to the personalization scheme. Additionally, all these methods are frequentist, specific to parametric models, and lack generalization bounds. For a comprehensive overview of PFL, please refer to Kulkarni et al. (2020).

**FL for probabilistic models.** Several FL methods aim to learn a global posterior at the server level, which is generally intractable. Accordingly, these studies resort to approximation techniques such as Laplace approximation (Al-Shedivat et al., 2020), Stein Variational Gradient Descent (SVGD) (Kassab and Simeone, 2022), and Variational Inference (VI) (Bui et al., 2018). In Corinzia and Buhmann (2019), personalized posteriors are achieved by training global and per-client variational factors

jointly, but this method faces scalability issues in systems with many clients since refining personalized models is performed sequentially. While this approach performs well in the tested scenarios, it is expected that the global variational factors cannot capture patterns among highly heterogeneous clients, and the framework becomes equivalent to clients learning solely on their local data, as emphasized by Marfoq et al. (2021). Recently, Zhang et al. (2022) introduced pFedBayes which is a bilevel optimization approach combining a server-level global prior with client-level personalized posteriors. In each round, clients calculate local posteriors by optimizing their data likelihood while adding regularization towards the global prior. The server calculates the prior by minimizing the average client loss. However, this approach, due to its bi-level optimization, imposes a significant computational burden. Additionally, it results in a heuristic posterior, forfeiting the Bayesian interpretation, due to the dependence of the prior on the client data. In contrast, we employ an inference framework, as proposed by Dziugaite and Roy (2018), to handle data-dependent priors. Akin to meta-PFL approaches, regularization towards a global model may limit the degree of personalization achievable.

The closest work to our approach is Achituve et al. (2021), which performs deep kernel learning for GPs by maximizing the average Log Marginal Likelihood (LML) across all clients, learning a shared prior, and regarding the posterior inference as personalization. Our method distinguishes itself from Achituve et al. (2021) by learning a distribution over priors instead of a single prior. Moreover, while Achituve et al. (2021) only considers zero-mean priors, our approach can learn both the mean and the covariance of the prior distribution. The reason for focusing on covariance learning and ignoring the mean in Achituve et al. (2021) may be that the LML implicitly regularizes the kernel, while the mean can quickly overfit, as suggested in Fortuin et al. (2020). Our approach overcomes this issue by incorporating a principled regularization term.

**PAC-Bayesian meta-learning**  The setup described in Section 1 is closely related to meta-learning within the PAC-Bayesian framework (Amit and Meir, 2018; Pentina and Lampert, 2014; Rothfuss et al., 2021). In this scheme, a meta-learner is presented with a sequence of heterogeneous learning tasks and aims to learn a hyper-posterior from previous tasks to facilitate posterior inference for a new unseen task. The meta-learner achieves this by establishing and minimizing a generalization bound on the new task using the previous tasks' data, establishing a data flow from existing to new tasks. While the mentioned approaches can be used for FL by treating each client as a task, they do not incentivize existing clients to collaborate in the scheme. In contrast, our approach focuses on the existing clients who are the primary concern of an FL system and minimizes a generalization bound on them. Particularly, our approach is an adaptation of Rothfuss et al. (2021) to PFL, where the key technical difference is dealing with *data-dependent* priors. Our obtained bounds are structurally similar to those of Rothfuss et al. (2021), allowing us to leverage similar methods for optimizing the hyper-posterior.

## 3  Preliminaries and notation

Consider $n \in \mathbb{N}$ existing clients, each characterized by an *unknown data distribution*, $\mathcal{D}_i$, that is over a support of $R_{\mathbf{z}} = R_{\mathbf{x}} \times R_y$. Every sample from $\mathcal{D}_i$, is a pair of features, $\mathbf{x}$, and the corresponding target, $y$, where $\mathbf{x} \in R_{\mathbf{x}} \subset \mathbb{R}^d$ and $y \in R_y \subset \mathbb{R}$. The support of features and targets, $R_{\mathbf{x}}$ and $R_y$, are identical among clients, and the target is assumed scalar for simplicity. Each client observes two distinct sets of $m_i \in \mathbb{N}$ and $\tilde{m}_i \in \mathbb{N} \cup \{0\}$ i.i.d samples from $\mathcal{D}_i$, denoted by $\mathcal{S}_i \sim \mathcal{D}_i^{m_i}$ and $\tilde{\mathcal{S}}_i \sim \mathcal{D}_i^{\tilde{m}_i}$. The first set, $\mathcal{S}_i$, collects the samples that client $i$ involves in both FL and personalization steps and is non-empty. The second set, $\tilde{\mathcal{S}}_i$, consists of the samples obtained later that are used only for model personalization and may be empty. We assume that $\tilde{m}_i \leq m_i$ for all $i$, and denote the number of clients with $\tilde{m}_i > 0$ by $n_2 \in \{0, \cdots, n\}$. We allow $\mathcal{D}_i \neq \mathcal{D}_j$ and $m_i \neq m_j$ to accommodate clients' heterogeneity. Still, we consider a distribution $\mathcal{T}$ over the unknown data distributions and the number of samples, $(\mathcal{D}_i, m_i) \sim \mathcal{T}$, to encode relatedness among clients. For notation convenience, we shall use $\mathcal{S} \coloneqq \{\mathcal{S}_i\}_{i=1}^n$, $\mathcal{D} \coloneqq \{\mathcal{D}_i\}_{i=1}^n$, $\mathbf{m} \coloneqq [m_1, \cdots, m_n]$, and $\tilde{\mathbf{m}} \coloneqq [\tilde{m}_1, \cdots, \tilde{m}_n]$.

Each client $i \in 1, \cdots, n$ aims to learn a hypothesis function $h_i : R_{\mathbf{x}} \to R_y$ that can be used for predicting the label $y_*$ of an unseen input $\mathbf{x}_*$ drawn from the unknown data distribution $\mathcal{D}_i$. The quality of $h$ at $(\mathbf{x}_*, y_*)$ is measured by a loss function which we assume to be bounded, $\ell : \mathcal{H} \times R_{\mathbf{z}} \to [a, b]$. Since the data distribution, $\mathcal{D}_i$, is unknown, standard methods select the *best* hypothesis according to the set of observed samples, $\mathcal{S}_i \cup \tilde{\mathcal{S}}_i$; for instance, by minimizing the empirical risk associated with $\ell$.

Using a single hypothesis based on limited observations leads to epistemic uncertainty (Draper, 1995) and overconfident target estimation (Kass and Raftery, 1995). To address this issue, the *PAC-Bayesian* framework (McAllester, 1999) places probability distributions over the hypothesis space, $\mathcal{H}$, and combines all possible hypotheses sampled from these distributions for making inferences. Two such distributions are enlisted: a *prior* distribution, $P$, and a *posterior* distribution, $Q_i$, which depends on the prior and the observations through a mapping, $Q_i := \mathbb{Q}(P, \mathcal{S}_i \cup \tilde{\mathcal{S}}_i)$. While the prior and posterior terms resemble the Bayesian terminology, the posterior mapping, $\mathbb{Q}$, is not (necessarily) obtained through Bayes' theorem. Additionally, recent advances in PAC-Bayesian analysis enable employing priors that depend *slightly* on the data to make the learning pipeline more data-driven, which is contrary to the Bayesian formalism (Rivasplata et al., 2020).

**Client-level PAC-Bayesian bounds**    The flexibility in choosing the mapping $\mathbb{Q}$ can be exploited to achieve desired characteristics. Ideally, clients aim to choose a $\mathbb{Q}$ that minimizes the *true risk*,

$$\mathcal{L}^C\big(\mathbb{Q}(P, \mathcal{S}_i \cup \tilde{\mathcal{S}}_i), \mathcal{D}_i\big) := \mathop{\mathbb{E}}_{h \sim \mathbb{Q}(P, \mathcal{S}_i \cup \tilde{\mathcal{S}}_i)} \mathop{\mathbb{E}}_{\mathbf{z} \sim \mathcal{D}_i} \big[\ell(h, \mathbf{z})\big], \tag{1}$$

where superscript $C$ indicates that (1) is calculated by a client. In most cases, $\mathcal{D}_i$ is unknown and the true loss is approximated by its empirical counterpart,

$$\hat{\mathcal{L}}^C\big(\mathbb{Q}(P, \mathcal{S}_i \cup \tilde{\mathcal{S}}_i), \mathcal{S}_i \cup \tilde{\mathcal{S}}_i\big) := \mathop{\mathbb{E}}_{h \sim \mathbb{Q}(P, \mathcal{S}_i \cup \tilde{\mathcal{S}}_i)} \Big[\frac{1}{m_i + \tilde{m}_i} \sum_{\mathbf{z} \in \mathcal{S}_i \cup \tilde{\mathcal{S}}_i} \ell(h, \mathbf{z})\Big]. \tag{2}$$

PAC-Bayes theory (McAllester, 1999) provides a guarantee on the worst loss clients may suffer by upper bounding the unknown true risk in (1) based on the empirical risk in (2). The original PAC bound by McAllester (1999) assumes that the prior, $P$, is independent of the data. However, in our setup, the prior is obtained from the FL algorithm acting on the data subset $\mathcal{S}_i$, hence, is *data-dependent*. Dziugaite and Roy (2018) propose a recipe for adapting any PAC bound with data-free prior to a data-dependent prior that is stable to slight changes in the data, formalized by the notion of *Differential Privacy (DP)* defined below.

**Definition 3.1** ((Dwork and Roth, 2014)). *Let $\epsilon_i \in \mathbb{R}_+$ and $\mathcal{A}$ be a randomized algorithm that takes a dataset as input and generates a stochastic output. The algorithm $\mathcal{A}$ is $\epsilon_i$-differentially private if for all datasets $\mathcal{S}_i$ and $\hat{\mathcal{S}}_i$ that differ in a single data sample and all subsets $\mathcal{O}$ of possible outcomes of $\mathcal{A}$,*

$$e^{-\epsilon_i} \, Pr[\mathcal{A}(\hat{\mathcal{S}}_i) \in \mathcal{O}] \leq Pr[\mathcal{A}(\mathcal{S}_i) \in \mathcal{O}] \leq e^{\epsilon_i} \, Pr[\mathcal{A}(\hat{\mathcal{S}}_i) \in \mathcal{O}], \tag{3}$$

*where the probability is taken over the randomness used by the algorithm.*

Intuitively, (3) states that $\mathcal{A}$ is stable w.r.t changing one sample of $\mathcal{S}$ (Dwork et al., 2015). The parameter $\epsilon$ controls the privacy/stability level and is smaller if $\mathcal{A}$ is more private/stable.

To state a PAC bound for client $i$, we regard the federated pipeline as a randomized algorithm that takes $\mathcal{S}_i$ as input and outputs $P = \mathcal{A}_{\mathcal{S} \setminus \mathcal{S}_i}(\mathcal{S}_i)$. The subscript $\mathcal{S} \setminus \mathcal{S}_i$ contains samples from all clients except $i$ and emphasizes that $P$ depends on data from other clients likewise. However, DP is only studied for the argument, $\mathcal{S}_i$. Randomization in $\mathcal{A}_{\mathcal{S} \setminus \mathcal{S}_i}$ arises from sampling the prior from the hyper-posterior.

When the prior is obtained through a differentially private algorithm, we apply the method of Dziugaite and Roy (2018) on a bound due to Alquier et al. (2016) to derive the following theorem.

**Theorem 3.1.** *Fix a data-dependent prior $P$ obtained through an $\epsilon_i$-DP algorithm, a data distribution $\mathcal{D}_i$, and a bounded loss function $\ell(\cdot, \cdot) \in [a, b]$. For every $\beta > 0$, confidence level $\delta \in (0, 1]$, and posterior $Q_i = \mathbb{Q}(P, \mathcal{S}_i \cup \tilde{\mathcal{S}}_i)$, the inequality*

$$\mathcal{L}^C\big(Q_i, \mathcal{D}_i\big) \leq \hat{\mathcal{L}}^C\big(Q_i, \mathcal{S}_i \cup \tilde{\mathcal{S}}_i\big) + \frac{1}{\beta}\Big(KL\big(Q_i \| P\big) + \frac{\beta^2 (b-a)^2}{8(m_i + \tilde{m}_i)} + I(\epsilon_i, m_i, \delta) + \ln\big(\frac{1}{\delta}\big)\Big) \tag{4}$$

*holds with probability at least $1 - \delta$ over $\mathcal{S}_i \sim \mathcal{D}_i^{m_i}$ and $\tilde{\mathcal{S}}_i \sim \mathcal{D}_i^{\tilde{m}_i}$. In the above, $I(\epsilon_i, m_i, \delta) = 0.5 m_i \epsilon_i^2 + \epsilon_i \sqrt{0.5 m_i \ln(4/\delta)} + \ln(2)$, and does not depend on the posterior. It is assumed that the KL divergence between $Q_i$ and $P$ exists and is denoted by $KL(Q_i \| P)$.*

*Remark.* The term $I(\epsilon_i, m_i, \delta)$ is the only change from the bound of Alquier et al. (2016) with a data-free prior.

We refer to Theorem 3.1 as the *client-level upper bound* because it provides a guarantee on the unknown true risk of each client $i$.

**Optimal posterior**  The bound in (4) holds for all $Q_i$ and thus can be minimized w.r.t $Q_i$ to obtain the tightest upper bound on the true risk. Since $\epsilon_i$ only reflects through the term $I$ in (4), the optimal posterior is the same as the minimizer of the bound with a data-free prior derived in Catoni (2007).

**Corollary 3.1.1** ((Catoni, 2007)). *Given a prior, $P$, obtained through an $\epsilon_i$-DP algorithm and observations, $\mathcal{S}_i \cup \tilde{\mathcal{S}}_i$, the optimal posterior minimizing the right-hand side of (4) is a Gibbs distribution:*

$$Q_i^* \coloneqq \mathbb{Q}^*(P, \mathcal{S}_i \cup \tilde{\mathcal{S}}_i) = P(h) \cdot \exp\Big(\frac{-\beta}{m_i + \tilde{m}_i} \sum_{\mathbf{z} \in \mathcal{S}_i \cup \tilde{\mathcal{S}}_i} \ell(h, \mathbf{z})\Big)/Z_\beta^C(P, \mathcal{S}_i \cup \tilde{\mathcal{S}}_i), \qquad (5)$$

*where $Z_\beta^C(P, \mathcal{S}_i \cup \tilde{\mathcal{S}}_i) \coloneqq \mathbb{E}_{h \sim P} \exp\Big(\frac{-\beta}{m_i + \tilde{m}_i} \sum_{\mathbf{z} \in \mathcal{S}_i \cup \tilde{\mathcal{S}}_i} \ell(h, \mathbf{z})\Big)$ is a normalization constant.*

Using the negative log-likelihood loss, $\ell(h, \mathbf{z}) \coloneqq -\ln \Pr[\mathbf{z}|h]$ and $\beta = m_i + \tilde{m}_i$, reduces $Z_\beta^C$ to the LML and the Gibbs posterior coincides with the Bayes posterior (Guedj, 2019).

Plugging in the closed-form formula of the optimal posterior (5) into the client-level bound obtains:

$$\mathcal{L}^C\big(\mathbb{Q}^*(P, \mathcal{S}_i \cup \tilde{\mathcal{S}}_i), \mathcal{D}_i\big) \leq \frac{1}{\beta}\Big(-\ln Z_\beta(P, \mathcal{S}_i \cup \tilde{\mathcal{S}}_i) + \frac{\beta^2(b-a)^2}{8(m_i + \tilde{m}_i)} + I(\epsilon_i, m_i, \delta) + \ln\big(\frac{1}{\delta}\big)\Big), \quad (6)$$

holding with probability at least $1 - \delta$ over $\mathcal{S}_i \sim \mathcal{D}_i^{m_i}, \tilde{\mathcal{S}}_i \sim \mathcal{D}_i^{\tilde{m}_i}$. The simplified bound (6) removes the explicit dependence on $Q_i$ and is tighter than the generic bound per (4).

In the rest of this paper, we assume that clients utilize $Q_i^*$ whenever the privacy requirement of Theorem 3.1 is satisfied. This is a mild assumption bearing into mind that the family of Gibbs posteriors (for different choices of $\ell$ and $\beta$) encompasses the Bayes posterior.

## 4    THEORETICAL FRAMEWORK FOR PAC-BAYESIAN FEDERATED LEARNING

**Overview of the approach.**  Having defined basic components, we reemphasize the main research problem. In this paper, we employ FL to select the prior distribution in a data-driven manner. With a similar justification of considering distributions over hypotheses, we formulate two distributions over priors: a *hyper-prior*, $\mathcal{P}$, and a *hyper-posterior*, $\mathcal{Q}$. The hyper-prior must be independent of observations, while the hyper-posterior depends on the observed samples by clients.

We assume the existence of a trusted server communicating with a set of existing clients, $1, \cdots, n$, each holding a set of observed samples, $\mathcal{S}_1, \cdots, \mathcal{S}_n$. The server aims to extract common knowledge in the form of a hyper-posterior distribution without directly accessing clients' datasets (as enforced by FL). The server communicates the learned hyper-posterior to clients, including those who participated in federated training and some new ones. The clients proceed by repeatedly drawing a prior from the received hyper-posterior and using it to calculate the optimal posterior per (5), involving any potential additional samples, $\tilde{\mathcal{S}}_i$ in the inference procedure. The goal is to find the *optimal hyper-posterior*, $\mathcal{Q}^*$, such that the posterior obtained through the described pipeline has a low true risk (1) for all clients. An overview of the setup is depicted in Fig. 1.

In the sequel, we consider hyper-posteriors, $\mathcal{Q}$, that satisfy the condition of having a finite $\epsilon \in \mathbb{R}_+$ such that sampling $P$ from $\mathcal{Q}$ preserves $\epsilon$-DP for all clients. This assumption enables us to use Corollary 3.1.1 and is rather weak as $\epsilon$ can be arbitrarily, albeit not infinitely, large. Analogous to the procedure in Section 3, we establish PAC bounds for the pair $\mathcal{P}$ and $\mathcal{Q}$. We derive a closed-form formula for the optimal hyper-posterior, $\mathcal{Q}^*$, which minimizes the PAC bound, and verify that $\mathcal{Q}^*$ satisfies the privacy assumption. Finally, we establish a PAC bound for new clients who sample their priors from $\mathcal{Q}^*$ without participating in the federated learning process. An interpretation through the principle of maximum entropy (Jaynes, 1957) and all proofs are provided in Appendices 8.1 and 8.2, respectively.

**Server-level PAC-Bayesian bound.**  Following the introduced inference setup, the quality of $\mathcal{Q}$ can be measured by *server-level true risk*, defined as:

$$\mathcal{L}^S(\mathcal{Q}, \mathcal{D}, \mathcal{S}, \tilde{\mathbf{m}}) \coloneqq \frac{1}{n} \sum_{i=1}^n \mathbb{E}_{P \sim \mathcal{Q}} \mathbb{E}_{\tilde{\mathcal{S}}_i \sim \mathcal{D}_i^{\tilde{m}_i}} \mathcal{L}^C(\mathbb{Q}^*(P, \mathcal{S}_i \cup \tilde{\mathcal{S}}_i), \mathcal{D}_i). \qquad (7)$$

Equation (7) averages the true loss across all clients, accounting for every possible set of new samples, $\tilde{\mathcal{S}}$, with sizes specified by $\tilde{\mathbf{m}}$. The data generating distributions $\mathcal{D}$ are unknown in the true loss definition per (7), calling for an empirical estimation merely based on the available information:

$$\hat{\mathcal{L}}^S(\mathcal{Q}, \mathcal{S}) := \frac{1}{n} \sum_{i=1}^{n} \mathop{\mathbb{E}}_{P \sim \mathcal{Q}} \hat{\mathcal{L}}^C(\mathbb{Q}^*(P, \mathcal{S}_i), \mathcal{S}_i), \tag{8}$$

where observed samples replace potential future data. We refer to (8) as the *server-level empirical loss* because it can be computed by the server by collecting summands from clients.

Below, we present our first main contribution, which is a PAC bound on server-level risks.

**Theorem 4.1.** *Let $\ell(\cdot, \cdot) \in [a, b]$ be a bounded loss function, $\beta \geq 1/n$,*

$$\Delta_i := \frac{1}{n} \min \left\{ b - a, b\left(e^{\frac{2\beta \tilde{m}_i}{m_i + \tilde{m}_i}(b-a)} - e^{\frac{-2\beta \tilde{m}_i}{m_i + \tilde{m}_i}(b-a)}\right) \right\} \quad , \quad \forall i \in \{1, \cdots, n\}.$$

*Assume clients employ the optimal posterior with parameter $\beta$. Let $\mathcal{Q}$ be a hyper-posterior such that sampling $P$ from $\mathcal{Q}$ preserves $\epsilon$-DP for all clients. For every hyper-prior $\mathcal{P}$ independent from $\mathcal{S}$, $\upsilon > 0$, $\lambda > n_2 + \upsilon$, and confidence level $\delta \in (0, 1)$,*

$$\mathcal{L}^S(\mathcal{Q}, \mathcal{D}, \mathcal{S}, \tilde{\mathbf{m}}) \leq \frac{-1}{n\beta} \sum_{i=1}^{n} \mathop{\mathbb{E}}_{P \sim \mathcal{Q}} \ln Z_\beta^C(P, \mathcal{S}_i) + \left(\frac{1}{n\beta} + \frac{n_2 + \upsilon}{\lambda}\right) KL(\mathcal{Q} \| \mathcal{P})$$

$$+ \frac{\beta(b-a)^2}{8n} \sum_{i=1}^{n} \frac{1}{m_i} + \frac{\lambda \sum_{i=1}^{n} \Delta_i^2}{8(n_2 + \upsilon)} + \frac{1}{\sqrt{n}} \ln\left(\frac{1}{\delta}\right), \tag{9}$$

*holds with probability at least $1 - \delta$ over $\mathcal{S}_i \sim \mathcal{D}_i^{m_i}$ and $\tilde{\mathcal{S}}_i \sim \mathcal{D}_i^{\tilde{m}_i}$ for $i = 1, \cdots, n$.*

The constant $\upsilon$ can be chosen very small and avoids numerical issues when $n_2 = 0$. The bound in (9) depends on the number of clients and the current dataset sizes, $n$ and $\mathbf{m}$. Additionally, it relies on a prediction of the number of clients who will have new samples and the corresponding number of new samples, $n_2$ and $\tilde{\mathbf{m}}$. While the server knows $n$ and $\mathbf{m}$, the estimates for $n_2$ and $\tilde{\mathbf{m}}$ might be coarse. The following lemma states that a pessimistic forecast results in a looser upper bound.

**Lemma 4.2.** *If the number of new samples of client $i$, $\tilde{m}_i \geq 0$, is unknown, Theorem 4.1 holds when replacing $\Delta_i$ with $(b - a)/n$ and counting client $i$ in $n_2$, i.e., as if $\tilde{m}_i > 0$.*

The asymptotic behavior and non-vacuousness of the client-level and server-level bounds are discussed in Appendix 8.3.1.

**Optimal hyper-posterior.** Our algorithm picks the *optimal hyper-posterior*, $\mathcal{Q}^*$, leading to the lowest upper bound on the server-level true risk per (4). Inspecting the structural similarity between the server and client-level bounds in (4) and (9), we arrive at a closed-form formula for $\mathcal{Q}^*$.

**Corollary 4.2.1.** *When clients use the optimal posterior, $Q_i^*$, the optimal hyper-posterior is a Gibbs distribution with parameter $\tau = \lambda / (\lambda + \beta n(n_2 + \upsilon))$:*

$$\mathcal{Q}^*(P) = \mathcal{P}(P) \cdot \exp\left(\tau \sum_{i=1}^{n} \ln\left(Z_\beta^C(P, \mathcal{S}_i)\right)\right) / Z_\tau^S(\mathcal{P}, \mathcal{S}),$$

*where $Z_\tau^S(\mathcal{P}, \mathcal{S}) := \mathbb{E}_{P \sim \mathcal{P}} \exp\left(\tau \sum_{i=1}^{n} \ln\left(Z_\beta^C(P, \mathcal{S}_i)\right)\right)$ is a normalization constant.*

The parameter $\tau$ depends on the number of clients from different types, $n$ and $n_2$, but not on the number of samples, $\mathbf{m}$ and $\tilde{\mathbf{m}}$. If $n_2$ is unknown, it can be replaced by counting $i$ in $n_2$, consistently with Lemma 4.2. In this case, a looser upper bound would be minimized.

The privacy of sampling a prior from $\mathcal{Q}^*$ is crucial to obtain $\epsilon_i$ for plugging it into the client-level bound (6) and for employing Theorem 4.1. We rely on a result by Mir (2012) that proves the DP of sampling from the Gibbs distribution.

**Lemma 4.3.** *A prior sampled from $\mathcal{Q}^*$ preserves $\epsilon_i$-DP for client $i$, where $\epsilon_i = 2\beta\tau(b - a)/m_i$.*

As a result, $\mathcal{Q}^*$ satisfies the privacy assumption of Theorem 4.1 with $\epsilon = \max_{i \in \{1, \cdots, n\}} \epsilon_i$. Complementary discussions regarding the role of DP in our framework are available in Appendix 8.4.

**PAC-Bayesian bound for new clients.** So far, we considered a fixed set of *existing* clients who participate in training the optimal hyper-posterior. In a realistic FL setup, there might be *new* clients (see $c4$) who join the system later and hence, do not engage in federated training. A new client holds a presumably small set of samples which leads to overfitting. Assuming the existing and new clients are similar, it is constructive for the new clients to readily employ $\mathcal{Q}^*$ without having contributed to training it. In this section, we establish a PAC bound for such new clients.

In Section 3, we introduced the distribution $\mathcal{T}$ to represent the similarity among the existing clients. Consistently, we expect that a new client $\iota$ is sampled from the same distribution, $(\mathcal{D}_\iota, \tilde{m}_\iota) \sim \mathcal{T}$. In line with the previous notation, $\tilde{m}_\iota$ is the cardinality of the set $\tilde{\mathcal{S}}_\iota \sim \mathcal{D}_\iota^{\tilde{m}_\iota}$ employed by client $\iota$ for personalization but held out from FL. We present below our second PAC-Bayesian bound.

**Lemma 4.4.** *For a new client $\iota$ sampled from $\mathcal{T}$ adopting $\mathbb{Q}^*$ and $\mathcal{Q}^*$ as per Corollaries 3.1.1, 4.2.1,*

$$\mathbb{E}_{(\mathcal{D}_\iota, \tilde{m}_\iota) \sim \mathcal{T}} \mathbb{E}_{\tilde{\mathcal{S}}_\iota \sim \mathcal{D}_\iota^{\tilde{m}_\iota}} \mathbb{E}_{P \sim \mathcal{Q}^*} \mathcal{L}^C\big(\mathbb{Q}^*(P, \tilde{\mathcal{S}}_\iota), \mathcal{D}_\iota\big) \leq -\big(\frac{1}{n\beta} + \frac{n_2 + \upsilon}{\lambda}\big) \ln Z_\tau^S(\mathcal{P}, \mathcal{S})$$
$$+ \frac{(b-a)^2}{8n}\big(\beta \sum_{i=1}^{n} \frac{1}{m_i} + \frac{\lambda}{n_2 + \upsilon}\big) + \frac{1}{\sqrt{n}} \ln(\frac{1}{\delta}),$$

*holds with probability at least $1 - \delta$ over $(\mathcal{D}_\iota, \tilde{m}_\iota) \sim \mathcal{T}$ and $\tilde{\mathcal{S}}_\iota \sim \mathcal{D}_\iota^{\tilde{m}_\iota}$.*

Since $\mathcal{Q}^*$ is tailored to minimize the server-level bound for existing clients, the bound in Lemma 4.4 is looser than that of Theorem 4.1 with $\mathcal{Q} = \mathcal{Q}^*$ (see Appendix 8.2.5 for the proof). This motivates the clients to actively engage in training $\mathcal{Q}^*$ rather than readily employing the learned hyper-posterior.

## 5 PRACTICAL FEDERATED IMPLEMENTATION

In Section 3, we motivated learning a distribution over priors when dealing with heterogeneous clients, where selecting a single best prior may not be accurate or feasible. Consequently, we derived the optimal hyper-posterior in Corollary 4.2.1. In this section, we address the computational limitations at both the client and server levels and propose a practical PFL algorithm.

**Models at the client level.** The formula for $\mathcal{Q}^*$ in Corollary 4.2.1 relies on $Z_\beta^C(P, \mathcal{S}_i)$ for $i \in \{0, \cdots, n\}$. As per Corollary 3.1.1, calculating $Z_\beta^C(P, \mathcal{S}_i)$ entails computing the expectation over all hypotheses sampled from the prior, which is in general intractable. We consider the negative log-likelihood loss and set $\beta = m_i + \tilde{m}_i$, which results in $Z_\beta$ aligning with the LML, as discussed in Section 3. We calculate the LML in two scenarios: when clients use Gaussian Processes (GPs) or Bayesian Neural Networks (BNNs). MLL is readily available in closed-form for GPs but is intractable for BNNs. To address this, we adopt the approximation method described in Rothfuss et al. (2021).

Inspired by Fortuin et al. (2020); Rothfuss et al. (2021), we parameterize the GP mean and kernel by two deep neural networks (NN) and consider a Gaussian likelihood. The resulting model enhances the expressive power and scalability to high-dimensional data of GPs (Wilson et al., 2016). Additional details about the models and likelihood formulas can be found in Appendices 8.3.2 and 8.3.3.

**SVGD at the server level.** Given $Z_\beta^C(P, \mathcal{S}_i)$, $\mathcal{Q}^*$ is computable up to the constant $Z_\tau^S(\mathcal{P}, \mathcal{S})$, which leaves sampling from $\mathcal{Q}^*$ intractable. Following Rothfuss et al. (2021), we use Stein Variational Gradient Descent (SVGD) (Liu and Wang, 2016) that approximates $\mathcal{Q}^*$ as a set of priors, $P_{\phi_1}, \cdots, P_{\phi_k}$. Each prior $P_{\phi_\kappa}$ is parameterized by $\phi_\kappa$. SVGD is initialized with a set of priors and then iteratively transports them to match $\mathcal{Q}^*$. This is achieved through a form of functional gradient descent on the SVGD loss (see Appendix 8.3.4), making it suitable for being integrated into an FL scheme. As SVGD is deterministic (Liu, 2017), the inherent privacy of $\mathcal{Q}^*$ established in Lemma 4.3 is compromised. To reintroduce privacy, a conventional approach involves injecting noise into the SVGD gradients (Geyer et al., 2017). We propose a privacy-preserving variant of PAC-PFL in Appendix 8.4.

**Federated algorithm.** PAC-PFL leverages the FedAvg (McMahan et al., 2016) technique to minimize the SVGD loss. Initially, we sample $P_{\phi_1}, \cdots, P_{\phi_k}$ from $\mathcal{P}$, which we define as a multi-variate Gaussian distribution with a diagonal covariance matrix. At each iteration, the server

randomly selects a subset of existing clients to send $\phi_1, \cdots, \phi_k$. This is sufficient to reconstruct $P_{\phi_1}, \cdots, P_{\phi_k}$. The selected clients compute the required SVGD gradients and send them back to the server. The server aggregates all gradients before using them to update the priors. A pseudocode is provided in Algorithm 1 with sub-routines *Client_Update* and *SVGD_Update* defined in Appendix 8.3.5. For a comprehensive list of parameters used in both our theoretical results and algorithm, along with selection guidelines, please refer to Appendix 8.3.6.

---

**Algorithm 1** PAC-PFL. Requires: number of SVGD priors ($k$), hyper-prior ($\mathcal{P}$), parameter ($\tau$), number of iterations ($T$), number of clients per iteration ($c$), mini-batch size ($b$), learning rate ($\eta$).

---
1: **Server executes:**
2:     Initialize priors $P_{\phi_1}, \ldots, P_{\phi_k} \overset{i.i.d}{\sim} \mathcal{P}^k$
3:     **for** $t = 1$ to $T$ **do**
4:         Select a random subset $\mathcal{C}_t$ of $c$ clients
5:         **for** each selected client $i$ in $\mathcal{C}_t$ **in parallel do**
6:             $\boldsymbol{G}_i \leftarrow$ *Client_Update*$(b, \phi_1, \ldots, \phi_k)$
7:         Aggregate gradients: $\boldsymbol{G} \leftarrow \frac{1}{c} \sum_{i \in \mathcal{C}_t} \boldsymbol{G}_i$
8:         Update priors: $\phi_1, \ldots, \phi_k \leftarrow$ *SVGD_Update*$(\eta, \tau, \boldsymbol{G}, \mathcal{P}, \phi_1, \ldots, \phi_k)$
9:     **return** SVGD approximation of $\mathcal{Q}^*$: $P_{\phi_1}, \ldots, P_{\phi_k}$

---

## 6 EXPERIMENTS

We empirically evaluate PAC-PFL on four datasets: a dataset of photovoltaic (PV) panels and a polynomial synthetic dataset for regression, alongside the FEMNIST (Caldas et al., 2019) and EMNIST (Cohen et al., 2017) datasets for classification. Our algorithm consistently outperforms federated and data-centric baselines, improving the prediction accuracy and the calibration of uncertainty estimates simultaneously. These enhancements are evident in the reduction of both variance and mean of these metrics across both existing and new clients.

**Datasets.** We consider a time-series dataset of PV generation from multiple houses within a city. Each house is a client, and the clients exhibit heterogeneity due to their diverse locations, variations in shadows, and distinct orientations relative to the sun. In this section, we investigate a highly heterogeneous scenario characterized by a bimodal distribution over the clients, where half of the clients are oriented toward the east and half toward the west. We consider 24 existing and 24 new clients, each having a small set of 150 training samples. We denote this scenario, involving challenges ($c$1-$c$4), as *PV-EW (150)*. Other setups for using the PV dataset are introduced in Appendix 8.5.

The FEMNIST dataset consists of handwritten characters from various writers, treated as clients, and we employ it for 10-way digit classification. The heterogeneity among clients arises from their distinct handwritings ($c$2). We select 40 clients and examine two scenarios: one with 20 samples per client and another with 500 samples per client on average. In the low-data case ($c$3), we demonstrate PAC-PFL's superior performance over all baselines. In the full-data case, we highlight the scalability of our algorithm with large datasets. The polynomial and the EMNIST datasets are detailed in Appendix 8.5.

**Baselines.** We compare PAC-PFL against *pFedGP* (Achituve et al., 2021) and *pFedBayes* (Zhang et al., 2022), two probabilistic PFL methods. Additionally, we include two non-federated approaches: *Vanilla*, where each client trains a model individually, and *Pooled*, where a single model is trained on a pooled dataset comprising data from all clients in a data-centric manner. The performance of the Pooled approach is expected to be low for heterogeneous clients due to the absence of personalization. Furthermore, we assess two frequentist PFL approaches for training NNs, namely *MAML* (Fallah et al., 2020) and *MTL* (Evgeniou and Pontil, 2004). Hyper-parameters for each method are tuned through cross-validation. Further baseline details can be found in Appendix 8.6.

**Metrics.** We assess prediction accuracy and calibration for each client. In regression tasks, we use *root standardized mean squared error* (RSMSE) that normalizes RMSE by the standard deviation of targets. For classification, we measure the percentage of correctly classified samples. Additionally, we compute the *calibration error* (CE), which quantifies the deviation of predicted confidence intervals

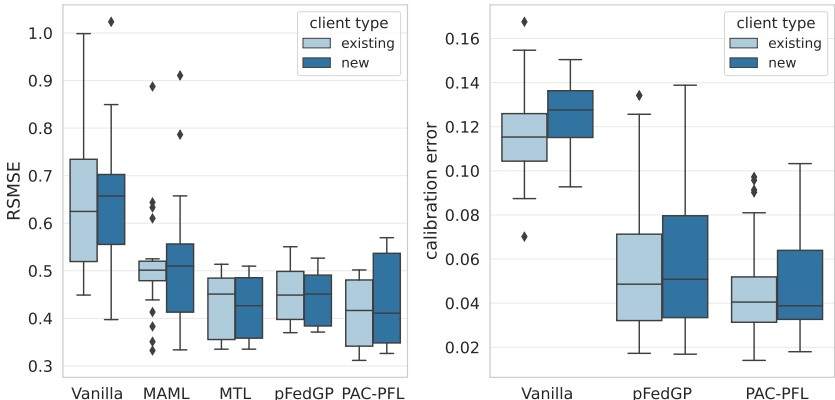

Figure 2: Box plots of test RSMSE and CE for existing and new clients in the *PV-EW (150)* dataset. The middle line in each box is the median. PAC-PFL consistently outperforms all baselines in terms of CE median, CE spread, and RSMSE median. The RSMSE spread is comparable to MTL and pFedGP.

from actual proportions of test data within those intervals (Kuleshov et al., 2018), using the formula in Rothfuss et al. (2021). The concept of CE applies exclusively to probabilistic models and is not relevant to frequentist baselines such as MAML and MTL. In evaluating FL methods, the sample mean of a metric is often a biased estimate due to correlations across clients. We utilize box plots to analyze the distribution of metric values across clients, offering a more comprehensive understanding.

**Results**   The results for *PV-EW (150)* are presented in Fig. 2, highlighting several key findings. MAML does not perform well, likely because a single gradient descent step lacks the necessary personalization ($c2$), while MTL and pFedGP showcase their capability to capture heterogeneity. The limited sample size ($c3$) adversely impacts the performance of the local Vanilla GP. Notably, PAC-PFL surpasses all baselines, leveraging its robust personalization capability to address ($c2$) and intrinsic regularization to resolve ($c3$). Notably, our method outperforms all baselines due to its strong personalization capability, addressing ($c2$), and inherent regularization, resolving ($c3$). Pooled GP results are not plotted due to poor performance but are reported in Appendix 8.6.

On the FEMNIST dataset with 20 samples per client, PAC-PFL archives a remarkable test accuracy of $94.2\%$ for existing clients, outperforming other baselines: pFedGP ($83.6\%$), pFedBayes ($87.0\%$), and FedAvg ($88.1\%$). Complementary experimental results are reported in Appendix 8.6.

## 7   CONCLUSION

This paper presents PAC-PFL, a novel PFL algorithm that enables the learning of probabilistic models. The proposed approach learns a shared hyper-posterior in a federated manner, which clients use to sample their priors for personalized posterior inference. To prevent overfitting, PAC-PFL minimizes an upper bound on the true risk of the clients participating in federated training. Moreover, the learned hyper-posterior can be applied to new clients who did not participate in the training, resulting in positive transfer. Conducting experiments on a heterogeneous dataset of PV panels and on the FEMNIST dataset, we empirically demonstrate that PAC-PFL produces accurate and well-calibrated predictions.

There are two main directions for future research: improving client-level computational complexity (detailed in Appendix 8.3.7) and addressing the privacy-utility trade-off more effectively. Our framework leverages DP to derive valid generalization bounds despite having data-dependent priors and to avoid data leakage, as typical in FL. While our theoretical results in Section 4 show that our ideal pipeline provides DP, we forfeit this property due to the SVGD approximation technique. We empirically validated that PAC-PFL avoids overfitting, though the bounds might not hold. DP can be reintroduced to prevent data leakage using the common method of injecting noise during training (Geyer et al., 2017), as demonstrated in Appendix 8.4, but this may compromise accuracy (Bagdasaryan et al., 2019). Exploring alternative privacy techniques is an avenue for future research.

## REPRODUCIBILITY STATEMENT

For all theorems and theoretical results, we present detailed assumptions and proofs in Appendix 8.2. Moreover, we provide a comprehensive table containing all parameters utilized throughout the paper in Appendix 8.3.6. This table highlights the interconnections between these parameters and marks the free parameters that can be tuned for optimal utilization of our algorithm.

Regarding the datasets, we employ the FEMNIST dataset, which is curated and maintained by the LEAF project (Caldas et al., 2019). We utilize the original train-test split provided with the data, without any additional preprocessing. The PV dataset can be accessed via the following link: `https://drive.google.com/drive/folders/153MeAlntN4VORHdgYQ3wG3OylW0SlBf9?usp=sharing`.

The source code for our PAC-PFL implementation using GP is accessible within the same Google Drive repository. Upon acceptance, we intend to make the source code for BNN publicly available. To facilitate the use of our software, we have incorporated a demonstration Jupyter Notebook in the source code repository. Furthermore, we have included pre-trained models for PAC-PFL and other baseline models for the PV dataset. Finally, we provide a notebook that generates the figures featured in the paper.

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

## 8 APPENDIX

### 8.1 INTERPRETATION THROUGH THE PRINCIPLE OF MAXIMUM ENTROPY

In this section, we provide additional justification for the optimal hyper-posterior derived in Corollary 4.2.1 based on the principle of maximum entropy (Jaynes, 1957). The principle of maximum entropy suggests that when only the class of a distribution is known, the distribution with the highest entropy should be chosen as the least-informative default. The distribution class can be specified by certain moment constraints.This principle is motivated by two key reasons: first, maximizing entropy minimizes the amount of prior information embedded in the distribution, allowing for a more agnostic representation; second, it aligns with the observation that many physical systems tend to evolve towards configurations of maximal entropy over time. While the principle of maximum entropy is commonly employed to derive prior probability distributions in Bayesian inference (Merwe and Skilling, 2010), we utilize it in the context of obtaining the optimal hyper-posterior distribution.

Consider the following constrained maximum entropy problem:

$$\max_{\mathcal{Q}} \; H_{\mathcal{P}}(\mathcal{Q}) \tag{10a}$$

$$s.t. \; -\mathbb{E}_{P \sim \mathcal{Q}} \ln Z_\beta(P, \mathcal{S}_i) + I(\epsilon_i, m_i, \delta) \leq -\mathbb{E}_{P \sim \mathcal{P}} \ln Z_\beta(P, \mathcal{S}_i) \quad \forall i \in \{1, \cdots, n\}. \tag{10b}$$

In Equation (10a), $H_{\mathcal{P}}(\mathcal{Q}) = -KL(\mathcal{Q}\|\mathcal{P})$ represents Jaynes' entropy with the hyper-prior $\mathcal{P}$ serving as the *invariant measure* (Jaynes, 1957). The objective is to maximize this entropy subject to $n$ constraints on the expectations of $\ln Z_\beta(P, \mathcal{S}_i)$ under the distribution $\mathcal{Q}$, as given in (10b). The constants $\epsilon_i$ and $I(\epsilon_i, m_i, \delta)$ are determined by Lemma 4.3 and Theorem 3.1, respectively. Below, we establish a connection between the maximum entropy problem and our approach in Section 4.

**Proposition 1.** *The minimizer of the server-level upper bound, $\mathcal{Q}^*$, derived in Corollary 4.2.1, coincides with the maximizer of the constrained maximum entropy problem presented in (10a)-(10b), when the maximum entropy problem is solved by optimizing the Lagrange function:*

$$\arg\max_{\mathcal{Q}} \; H_{\mathcal{P}}(\mathcal{Q}) + \tau \sum_{i=1}^{n} \Big( \mathbb{E}_{P \sim \mathcal{Q}} \ln Z_\beta(P, \mathcal{S}_i) - I(\epsilon_i, m_i, \delta) - \mathbb{E}_{P \sim \mathcal{P}} \ln Z_\beta(P, \mathcal{S}_i) \Big), \tag{11}$$

*where the Lagrange multiplier $\tau$ is used for all constraints. The constant $\tau$ is as per Corollary 4.2.1.*

*Proof.* Let $\tilde{\mathcal{Q}}^*$ denote the maximizer of the Lagrange function in (11). By removing the terms in (11) that are constant with respect to $\mathcal{Q}$, we obtain:

$$\tilde{\mathcal{Q}}^* = \arg\max_{\mathcal{Q}} \; H_{\mathcal{P}}(\mathcal{Q}) + \tau \sum_{i=1}^{n} \mathbb{E}_{P \sim \mathcal{Q}} \ln Z_\beta(P, \mathcal{S}_i). \tag{12}$$

According to the definition of Jaynes' entropy,

$$\tilde{\mathcal{Q}}^* = \arg\min_{\mathcal{Q}} \; \frac{1}{\tau} KL(\mathcal{Q}\|\mathcal{P}) - \sum_{i=1}^{n} \mathbb{E}_{P \sim \mathcal{Q}} \ln Z_\beta(P, \mathcal{S}_i), \tag{13}$$

where we have multiplied the objective in (12) by $-1/\tau$ and changed maximization into minimization. By substituting the formula for $\tau$, one can verify that (13) is equivalent to the server-level upper bound, except for some constant values. Hence, $\tilde{\mathcal{Q}}^* = \mathcal{Q}^*$. □

The maximum entropy interpretation of $\mathcal{Q}^*$ allows us to analyze the effect of sampling $P$ from $\mathcal{Q}^*$ on client-level bounds. If client $i$ chooses not to participate in FL and decides not to use $\mathcal{Q}^*$, the best alternative approach is to sample $P$ from $\mathcal{P}$. The following corollary provides a comparison between sampling $P$ from $\mathcal{Q}^*$ and sampling $P$ from $\mathcal{P}$.

**Corollary 8.0.1.** *The $i$-th constraint in (10b) imposes that the expected upper bound for client $i$ is tighter when sampling the prior $P$ from the optimal hyper-posterior, $\mathcal{Q}^*$, compared to sampling from the hyper-prior, $\mathcal{P}$.*

*Proof.* Since a prior which is sampled from the hyper-prior is no longer data-dependent, we utilize the result from Alquier et al. (2016) to derive a bound for client $i$:

$$\mathcal{L}^C(Q_i, \mathcal{D}_i) \leq \hat{\mathcal{L}}^C(Q_i, \mathcal{S}_i) + \frac{1}{\beta}\Big(KL(Q_i\|P) + \frac{\beta^2(b-a)^2}{8m_i} + \ln\big(\frac{1}{\delta}\big)\Big), \tag{14}$$

which holds with probability at least $1 - \delta$ over $\mathcal{S}_i \sim \mathcal{D}_i^{m_i}$. The upper bound in (14) is equal to the upper bound in (4) with $\tilde{\mathcal{S}}_i = \emptyset$, except for the constant term $I$. Therefore, the posterior that minimizes the right-hand side of (14) is the same as $\mathbb{Q}^*(P, \mathcal{S}_i)$ derived in Corollary 3.1.1 with $\tilde{\mathcal{S}}_i = \emptyset$. By plugging $\mathbb{Q}^*(P, \mathcal{S}_i)$ into (14), we obtain the counterpart of (6):

$$\mathcal{L}^C(\mathbb{Q}^*(P, \mathcal{S}_i), \mathcal{D}_i) \leq \frac{1}{\beta}\Big(-\ln Z_\beta(P, \mathcal{S}_i) + \frac{\beta^2(b-a)^2}{8m_i} + \ln\big(\frac{1}{\delta}\big)\Big), \tag{15}$$

holding with probability at least $1 - \delta$ over $\mathcal{S}_i \sim \mathcal{D}_i^{m_i}$. The $i$-th constraint in (10b) is obtained by taking the expectation of the upper bounds in (6) and (15) when $P \sim \mathcal{Q}^*$ and $P \sim \mathcal{P}$, respectively, and enforcing that the former is smaller than the latter. □

According to Corollary 8.0.1, participating in FL is beneficial for client $i$ if the $i$-th constraint is satisfied. However, since we solved the constrained problem per (10a)-(10b) using the Lagrange method with a single multiplier, the constraints might be violated. The constraints are more likely to be satisfied when $\tau$ is large, which can be achieved by having small $n$, $n_2$, and $\beta$, while simultaneously having a large value for $\lambda_E$. In other words, when there are fewer clients participating in FL, the hyper-posterior is more likely to provide improvements for those clients.

*Remark.* In a similar manner, we can avoid any PAC arguments and use the principle of minimum cross entropy for calculating the posterior. With these two rules, the optimal posterior and hyper-posterior are the same as those obtained by minimizing client-level and server-level PAC bounds.

## 8.2 PROOFS AND DERIVATIONS

We first mention without proof a powerful lemma, called the change of measure inequality, that is the basis of proving PAC bounds in most papers. The statement below is adapted from the Appendix of Pentina and Lampert (2014).

**Lemma 8.1** ((Pentina and Lampert, 2014)). *Let $f$ be a random variable taking values in a set $A$ and let $X_1, \cdots, X_l$ be $l$ independent random variables with each $X_k$ distributed according to $\mu_k$ over the set $A_k$. For functions $g_k : A \times A_k \rightarrow \mathbb{R}$, $k = 1, \cdots, l$, let $\xi_k(f) = \underset{X_k \sim \mu_k}{\mathbb{E}} g_k(f, X_k)$ denote the expectation of $g_k$ under $X_k \sim \mu_k$ as a function of $f$. Then, for any fixed distributions $\pi, \rho$ over $A$ and any $\gamma > 0$, we have that*

$$\underset{f \sim \rho}{\mathbb{E}}\Big[\sum_{k=1}^{l}\big(\xi_k(f) - g_k(f, X_k)\big)\Big] \leq \frac{1}{\gamma}KL(\rho\|\pi) + \frac{1}{\gamma}\psi(\gamma),$$

*where $\psi(\gamma) := \ln \mathbb{E}_{f \sim \pi}\Big[e^{\gamma \sum_{k=1}^{l}\big(\xi_k(f) - g_k(f, X_k)\big)}\Big]$ is referred to as the log moment-generating function.*

When $\xi_k - g_k$ is bounded, we bound the expectation of the log moment-generating function in Corollary 8.2.1, which uses the Hoeffding's lemma stated below.

**Lemma 8.2** ((Hoeffding, 1963)). *Let $Y$ be a zero-mean real-valued random variable such that $Y \in [a, b]$ almost surely, i.e. with probability one. Then for any $\gamma > 0$:*

$$\mathbb{E}\big[e^{\gamma Y}\big] \leq e^{\frac{\gamma^2}{8}(b-a)^2}.$$

**Corollary 8.2.1.** *If $\xi_k(f) - g_k(f, X_k) \in [a_k, b_k]$ almost surely for all $f$ and $X_k$, it holds for every $\gamma \geq 1$ that $\mathbb{E}_{X_1 \sim \mu_1} \cdots \mathbb{E}_{X_l \sim \mu_l} e^{\frac{1}{\gamma}\psi(\gamma)} \leq e^{\frac{\gamma}{8}\sum_{k=1}^{l}(b_k - a_k)^2}.$*

*Proof of Corollary 8.2.1.* By taking the expectation of the moment-generating function w.r.t every $X_k$,

$$\mathop{\mathbb{E}}_{X_1 \sim \mu_1} \cdots \mathop{\mathbb{E}}_{X_l \sim \mu_l} e^{\psi(\gamma)} = \mathop{\mathbb{E}}_{X_1 \sim \mu_1} \cdots \mathop{\mathbb{E}}_{X_l \sim \mu_l} \mathop{\mathbb{E}}_{f \sim \pi} \Big[ \prod_{k=1}^{l} e^{\gamma \left( \xi_k(f) - g_k(f, X_k) \right)} \Big]$$

$$= \mathop{\mathbb{E}}_{f \sim \pi} \mathop{\mathbb{E}}_{X_1 \sim \mu_1} \cdots \mathop{\mathbb{E}}_{X_l \sim \mu_l} \Big[ \prod_{k=1}^{l} e^{\gamma \left( \xi_k(f) - g_k(f, X_k) \right)} \Big],$$

where in the last line we have changed the order of expectations. For a given $f$, the terms $\xi_k(f) - g_k(f, X_k)$ for $k \in \{1, \cdots, l\}$ are independent from each other which allows applying Lemma 8.2:

$$\mathop{\mathbb{E}}_{X_1 \sim \mu_1} \cdots \mathop{\mathbb{E}}_{X_l \sim \mu_l} e^{\psi(\gamma)} = \mathop{\mathbb{E}}_{f \sim \pi} \Big[ \prod_{k=1}^{l} \mathop{\mathbb{E}}_{X_k \sim \mu_k} e^{\gamma \left( \xi_k(f) - g_k(f, X_k) \right)} \Big]$$

$$\leq \mathop{\mathbb{E}}_{f \sim \pi} \Big[ \prod_{k=1}^{l} e^{\frac{\gamma^2 (b_k - a_k)^2}{8}} \Big] = e^{\frac{\gamma^2}{8} \sum_{k=1}^{l} (b_k - a_k)^2}.$$

Since $1/\gamma \leq \gamma$, we can use Jensen's inequality (Jensen, 1906) to write:

$$\mathop{\mathbb{E}}_{X_1 \sim \mu_1} \cdots \mathop{\mathbb{E}}_{X_l \sim \mu_l} e^{\frac{1}{\gamma} \psi(\gamma)} \leq e^{\frac{\gamma}{8} \sum_{k=1}^{l} (b_k - a_k)^2}.$$

$\square$

One can utilize Markov's inequality[1] to remove the expectation in Corollary 8.2.1 and obtain a probabilistic bound on $\psi(\gamma)$. We will use Corollary 8.2.1 multiple times, and thus, postpone applying Markov's inequality to avoid simultaneous stochastic inequalities which must be combined with a union bound argument.

### 8.2.1 PROOF OF THEOREM 4.1

The proof of Theorem 4.1 is carried out in three steps. The first two steps bound the true risk of clients with $\tilde{S}_i = \varnothing$ and clients with enlarged datasets, respectively. By merging these two, we will obtain a bound on the server-level true risk. In the last step, the closed-form formula of the optimal posterior is exploited to make some simplifications. The main assumptions are independence of clients (guaranteeing independence of $X_k$ in Lemma 8.1), boundedness of the loss function (needed to apply Corollary 8.2.1), adoption of the optimal posterior (yielding the bound per (6)), and existence of a finite $\epsilon \in \mathbb{R}_+$ such that sampling $P$ from $\mathcal{Q}$ preserves $\epsilon$-DP for all clients (required to use Corollary 3.1.1).

**Step 1.** We apply Lemma 8.1 with the following instances: take $l = \sum_{i=1}^{n} m_i$ and assign a random variable to each observed sample by clients, $X_k = \mathbf{z}_{ij}$, where $\mathbf{z}_{ij}$ is the $j$'th sample of $\mathcal{S}_i$. Let $\alpha : \{1, \cdots, l\} \to \{1, \cdots, n\}$ be a mapping from each random variable $X_k$ to the corresponding client, $\alpha(k) = i$ if $X_k = \mathbf{z}_{ij}$ for some $j$. Correspondingly, we take $\mu_k = \mathcal{D}_{\alpha(k)}$ to be the respective distribution. Further, we set $f = (P, h_1, \cdots, h_n)$ to be a tuple of one prior and $n$ hypotheses and consider distributions $\pi = (\mathcal{P}, P, \cdots, P)$ and $\rho = \big( \mathcal{Q}, \mathbb{Q}^*(P, \mathcal{S}_1), \cdots, \mathbb{Q}^*(P, \mathcal{S}_n) \big)$ over it. Each function $g_k$ is designated to be one of the summands in the empirical server-level risk, $g_k(f, X_k) = \frac{1}{nm_{\alpha(k)}} \ell(h_{\alpha(k)}, X_k)$. By invoking Lemma 8.1 with $\gamma = \lambda_1 \geq 1$, we have:

$$\frac{1}{n} \mathop{\mathbb{E}}_{P \sim \mathcal{Q}} \sum_{i=1}^{n} \mathcal{L}^C \big( \mathbb{Q}^*(P, \mathcal{S}_i), \mathcal{D}_i \big) \leq \frac{1}{n} \mathop{\mathbb{E}}_{P \sim \mathcal{Q}} \sum_{i=1}^{n} \hat{\mathcal{L}}^C \big( \mathbb{Q}^*(P, \mathcal{S}_i), \mathcal{S}_i \big) \tag{$*$}$$

$$+ \frac{1}{\lambda_1} KL \big( \mathcal{Q} \| \mathcal{P} \big) + \frac{1}{\lambda_1} \sum_{i=1}^{n} \mathop{\mathbb{E}}_{P \sim \mathcal{Q}} KL \big( \mathbb{Q}^*(P, \mathcal{S}_i) \| P \big) \tag{$\dagger$}$$

$$+ \frac{1}{\lambda_1} \psi_1(\lambda_1). \tag{16}$$

---

[1] According to Markov's inequality, if $X$ is a nonnegative random variable and $a > 0$, then $\Pr[X \geq a] \leq \mathbb{E}[X]/a$.

Line (†) is equal to $KL(\rho\|\pi)$ due to (13-15) in Rothfuss et al. (2021), and $\psi_1(\lambda_1)$ is a log moment-generating function defined as:

$$\psi_1(\lambda_1) := \ln \mathop{\mathbb{E}}_{P\sim\mathcal{P}} \mathop{\mathbb{E}}_{h\sim P} e^{\frac{\lambda_1}{n}\sum_{i=1}^n \left(\mathbb{E}_{\mathbf{z}\sim\mathcal{D}_i}\ell(h,\mathbf{z})-\frac{1}{m_i}\sum_{\mathbf{z}\in\mathcal{S}_i}\ell(h,\mathbf{z})\right)}.$$

For $\ell \in [a,b]$, we apply Corollary 8.2.1 acknowledging that $|\xi_k - g_k| \leq (b-a)/nm_{\alpha(k)}$, hence, obtaining:

$$\mathop{\mathbb{E}}_{\mathcal{S}_1\sim\mathcal{D}_1^{m_1}} \cdots \mathop{\mathbb{E}}_{\mathcal{S}_n\sim\mathcal{D}_n^{m_n}} e^{\frac{1}{\lambda_1}\psi_1(\lambda_1)} \leq e^{\frac{\lambda_1}{8n^2}(b-a)^2\sum_{i=1}^n \frac{1}{m_i}}. \tag{17}$$

The right-hand side of (*) matches the server-level empirical loss per (8), but the left-hand side is different from the true risk in (7), as new clients and new samples are missing.

**Step 2.** In the second step, we use $l = n$ and assign one random variable $X_k$ to each client. To maintain a cohesive notation, we will use subscript $i$ instead of $k$ for elements of Lemma 8.1. We substitute each $X_i$ with a subset of $\tilde{m}_i$ samples from $\mathcal{S}_i$ drawn without replacement, which is possible as $\tilde{m}_i \leq m_i$. Set $f = P$, $\pi = \mathcal{P}$, $\rho = \mathcal{Q}$, and $g_i(f, X_i) = \frac{1}{n}\mathcal{L}^C(\mathbb{Q}^*(f,\mathcal{S}_i\cup X_i),\mathcal{D}_i)$. Notice that $\mathcal{S}_i$ and $\mathcal{D}_i$ are embedded in the definition of $g_i$ and not given as function arguments. Let $\lambda_2 = \lambda/(n_2 + \upsilon) \geq 1$, where $n_2$ is the number of clients with $\tilde{m}_i > 0$ and $\upsilon$ is a small positive number. By applying Lemma (8.1) with parameter $\gamma = \lambda_2$, we obtain:

$$\frac{1}{n}\mathop{\mathbb{E}}_{P\sim\mathcal{Q}}\sum_{i=1}^n \mathop{\mathbb{E}}_{\tilde{\mathcal{S}}_i\sim\mathcal{D}_i^{\tilde{m}_i}} \mathcal{L}^C(\mathbb{Q}^*(P,\mathcal{S}_i\cup\tilde{\mathcal{S}}_i),\mathcal{D}_i) \leq \frac{1}{n}\mathop{\mathbb{E}}_{P\sim\mathcal{Q}}\sum_{i=1}^n \mathcal{L}^C(\mathbb{Q}^*(P,\mathcal{S}_i),\mathcal{D}_i) \tag{$\diamond$}$$

$$+ \frac{1}{\lambda_2}KL(\mathcal{Q}\|\mathcal{P}) + \frac{1}{\lambda_2}\psi_2(\lambda_2), \tag{18}$$

$$\psi_2(\lambda_2) := \ln \mathop{\mathbb{E}}_{P\sim\mathcal{P}} e^{\frac{\lambda_2}{n}\sum_{i=1}^n \left(\mathbb{E}_{\tilde{\mathcal{S}}_i\sim\mathcal{D}_i^{\tilde{m}_i}}\mathcal{L}^C\left(\mathbb{Q}^*(P,\mathcal{S}_i\cup\tilde{\mathcal{S}}_i),\mathcal{D}_i\right)-\mathcal{L}^C\left(\mathbb{Q}^*(P,\mathcal{S}_i),\mathcal{D}_i\right)\right)}.$$

When $\tilde{m}_i = 0$ for all clients, the two sides of ($\diamond$) are equal, $\psi_2(\lambda_2)$ is zero, and the weight of the KL term in (18) goes to zero as $\upsilon \to 0$. Thus, by setting $\lambda_2$ proportional to $n_2 + \upsilon$, (16) and (18) are consistent.

It is evident from the definition of $\xi_i$ and $g_i$ that $|\xi_i - g_i| \leq (b-a)/n$. Still, we can exploit the explicit formula of $\mathbb{Q}^*$ per (5) to obtain a potentially smaller range by controlling the effect of a new sample of size $\tilde{m}_i$ on the optimal posterior. Let $Q_i^* = \mathbb{Q}^*(P,\mathcal{S}_i)$ and $\tilde{Q}_i^* = \mathbb{Q}^*(P,\mathcal{S}_i\cup\tilde{\mathcal{S}}_i)$ for some fixed $P$ and $\tilde{\mathcal{S}}_i$. From the closed form formula of $\mathbb{Q}^*$ per (5), for every $h \in \mathcal{H}$:

$$\frac{\tilde{Q}_i^*(h)}{Q_i^*(h)} = \frac{e^{\frac{-\beta}{m_i+\tilde{m}_i}\sum_{\mathbf{z}\in\mathcal{S}_i\cup\tilde{\mathcal{S}}_i}\ell(h,\mathbf{z})}}{e^{\frac{-\beta}{m_i}\sum_{\mathbf{z}\in\mathcal{S}_i}\ell(h,\mathbf{z})}} \cdot \frac{\mathbb{E}_{h\sim P} e^{\frac{-\beta}{m_i}\sum_{\mathbf{z}\in\mathcal{S}_i}\ell(h,\mathbf{z})}}{\mathbb{E}_{h\sim P} e^{\frac{-\beta}{m_i+\tilde{m}_i}\sum_{\mathbf{z}\in\mathcal{S}_i\cup\tilde{\mathcal{S}}_i}\ell(h,\mathbf{z})}} \tag{19}$$

From the bounded loss assumption, we have:

$$\left|\frac{1}{m_i+\tilde{m}_i}\sum_{\mathbf{z}\in\mathcal{S}_i\cup\tilde{\mathcal{S}}_i}\ell(h,\mathbf{z}) - \frac{1}{m_i}\sum_{\mathbf{z}\in\mathcal{S}_i}\ell(h,\mathbf{z})\right| \leq \frac{\tilde{m}_i(b-a)}{m_i+\tilde{m}_i}. \tag{20}$$

From (19) and (20), $\frac{\tilde{Q}_i^*(h)}{Q_i^*(h)} \in \left[e^{\frac{-2\beta\tilde{m}_i}{m_i+\tilde{m}_i}(b-a)}, e^{\frac{2\beta\tilde{m}_i}{m_i+\tilde{m}_i}(b-a)}\right]$, which obtains a bound on $\xi_i(f) - g_i(f,X_i)$:

$$\xi_i(f) - g_i(f,X_i) = \frac{1}{n}\mathop{\mathbb{E}}_{\tilde{\mathcal{S}}_i\sim\mathcal{D}_i^{\tilde{m}_i}} \mathcal{L}^C(\mathbb{Q}^*(P,\mathcal{S}_i\cup\tilde{\mathcal{S}}_i),\mathcal{D}_i) - \frac{1}{n}\mathcal{L}^C(\mathbb{Q}^*(P,\mathcal{S}_i),\mathcal{D}_i)$$

$$= \frac{1}{n}\mathop{\mathbb{E}}_{\tilde{\mathcal{S}}_i\sim\mathcal{D}_i^{\tilde{m}_i}} \mathop{\mathbb{E}}_{\mathbf{z}\sim\mathcal{D}_i} \int_{\mathcal{H}} \ell(h,\mathbf{z})(\tilde{Q}_i^*(h) - Q_i^*(h))d_h$$

$$\in \frac{1}{n}\mathcal{L}^C(\mathbb{Q}^*(P,\mathcal{S}_i),\mathcal{D}_i)\cdot\left[e^{\frac{-2\beta\tilde{m}_i}{m_i+\tilde{m}_i}(b-a)} - 1, e^{\frac{2\beta\tilde{m}_i}{m_i+\tilde{m}_i}(b-a)} - 1\right]$$

$$\subset \frac{b}{n}\left[\left(e^{\frac{-2\beta\tilde{m}_i}{m_i+\tilde{m}_i}(b-a)} - 1\right), \left(e^{\frac{2\beta\tilde{m}_i}{m_i+\tilde{m}_i}(b-a)} - 1\right)\right]. \tag{21}$$

Comparing (21) and the naive inequality $|\xi_i - g_i| \leq (b-a)/n$, it can be verified that $|\xi_i - g_i| \leq \Delta_i$, where:

$$\Delta_i := \frac{1}{n} \min \left\{ b-a, b \left( e^{\frac{2\beta \tilde{m}_i}{m_i + \tilde{m}_i}(b-a)} - e^{\frac{-2\beta \tilde{m}_i}{m_i + \tilde{m}_i}(b-a)} \right) \right\}.$$

It is possible to obtain a tighter range, $\Delta_i < (b-a)/n$, when $m_i$ is large, $\tilde{m}_i$ is small, or $\beta$ is small. Particularly, for $\tilde{m}_i = 0$, $\Delta_i = 0$ but $b-a$ provides a vacuous bound.

By applying Corollary 8.2.1, one obtains:

$$\mathop{\mathbb{E}}_{\tilde{\mathcal{S}}_1 \sim \mathcal{D}_1^{\tilde{m}_1}} \cdots \mathop{\mathbb{E}}_{\tilde{\mathcal{S}}_n \sim \mathcal{D}_n^{\tilde{m}_n}} e^{1/\lambda_2 \psi_2(\lambda_2)} \leq e^{\frac{\lambda_2}{8} \sum_{i=1}^n \Delta_i^2} \leq e^{\frac{\lambda_2}{8n}(b-a)^2}. \tag{22}$$

Note that $\mathcal{S}_i \neq \varnothing$ is required for defining $X_i$. Hence, new clients are not added to the analysis yet.

**Merging steps 1 and 2.** Bringing (16) and (18) together, we get:

$$\mathcal{L}^S(\mathcal{Q}, \mathcal{D}, \mathcal{S}, \tilde{\mathbf{m}}) \leq \hat{\mathcal{L}}^S(\mathcal{Q}, \mathcal{S}) + \left( \frac{1}{\lambda_1} + \frac{1}{\lambda_2} \right) KL(\mathcal{Q} \| \mathcal{P}) + \frac{1}{\lambda_1} \sum_{i=1}^n \mathop{\mathbb{E}}_{P \sim \mathcal{Q}} KL\left( \mathbb{Q}^*(P, \mathcal{S}_i) \| P \right)$$
$$+ \frac{1}{\lambda_1} \psi_1(\lambda_1) + \frac{1}{\lambda_2} \psi_2(\lambda_2). \tag{23}$$

Equation (23) provides a bound on the server-level true loss and incorporates various components such as the empirical loss, complexity penalty terms (represented by KL divergences between the posterior and the prior, and between the hyper-posterior and the hyper-prior), and two log moment-generating functions.

Next, we bound the weighted sum of the log moment-generating functions in (23) when $\ell \in [a, b]$. One obtains from Markov's inequality that:

$$Pr\left[ e^{\frac{1}{\lambda_1} \psi_1(\lambda_1) + \frac{1}{\lambda_2} \psi_2(\lambda_2)} \leq \frac{1}{\delta} e^{\frac{\lambda_1 (b-a)^2}{8n^2} \sum_{i=1}^n \frac{1}{m_i} + \frac{\lambda_2}{8} \sum_{i=1}^n \Delta_i^2} \right] \geq$$
$$1 - \frac{\mathbb{E}_{\mathcal{S}_1 \sim \mathcal{D}_1^{m_1}} \cdots \mathbb{E}_{\mathcal{S}_n \sim \mathcal{D}_n^{m_n}} \mathbb{E}_{\tilde{\mathcal{S}}_1 \sim \mathcal{D}_1^{\tilde{m}_1}} \cdots \mathbb{E}_{\tilde{\mathcal{S}}_n \sim \mathcal{D}_n^{\tilde{m}_n}} e^{\frac{1}{\lambda_1} \psi_1(\lambda_1) + \frac{1}{\lambda_2} \psi_2(\lambda_2)}}{\frac{1}{\delta} e^{\frac{\lambda_1 (b-a)^2}{8n^2} \sum_{i=1}^n \frac{1}{m_i} + \frac{\lambda_2}{8} \sum_{i=1}^n \Delta_i^2}}, \tag{24}$$

where the probability is taken over $\mathcal{S}_i \sim \mathcal{D}_i^{m_i}$ and $\tilde{\mathcal{S}}_i \sim \mathcal{D}_i^{\tilde{m}_i}$ for $i = 1, \cdots, n$ and $\delta \in (0, 1)$ is the confidence level given in Theorem 4.1. To bound the right-hand side, we multiply (17) and (22), which is possible as they involve expectations over independent random variables, resulting in:

$$\mathop{\mathbb{E}}_{\mathcal{S}_1 \sim \mathcal{D}_1^{m_1}} \cdots \mathop{\mathbb{E}}_{\mathcal{S}_n \sim \mathcal{D}_n^{m_n}} \mathop{\mathbb{E}}_{\tilde{\mathcal{S}}_1 \sim \mathcal{D}_1^{\tilde{m}_1}} \cdots \mathop{\mathbb{E}}_{\tilde{\mathcal{S}}_n \sim \mathcal{D}_n^{\tilde{m}_n}} e^{\frac{1}{\lambda_1} \psi_1(\lambda_1) + \frac{1}{\lambda_2} \psi_2(\lambda_2)} \leq e^{\frac{\lambda_1 (b-a)^2}{8n^2} \sum_{i=1}^n \frac{1}{m_i} + \frac{\lambda_2}{8} \sum_{i=1}^n \Delta_i^2}. \tag{25}$$

By putting together (24) and (25), we derive:

$$Pr\left[ \frac{1}{\lambda_1} \psi_1(\lambda_1) + \frac{1}{\lambda_2} \psi_2(\lambda_2) \leq \frac{\lambda_1 (b-a)^2}{8n^2} \sum_{i=1}^n \frac{1}{m_i} + \frac{\lambda_2}{8} \sum_{i=1}^n \Delta_i^2 + \ln(\frac{1}{\delta}) \right] =$$
$$Pr\left[ e^{\frac{1}{\lambda_1} \psi_1(\lambda_1) + \frac{1}{\lambda_2} \psi_2(\lambda_2)} \leq \frac{1}{\delta} e^{\frac{\lambda_1 (b-a)^2}{8n^2} \sum_{i=1}^n \frac{1}{m_i} + \frac{\lambda_2}{8} \sum_{i=1}^n \Delta_i^2} \right] \geq 1 - \delta, \tag{26}$$

where the probability is taken over $\mathcal{S}_i \sim \mathcal{D}_i^{m_i}$ and $\tilde{\mathcal{S}}_i \sim \mathcal{D}_i^{\tilde{m}_i}$ for $i = 1, \cdots, n$.

One obtains from (23) and (26) that:

$$Pr\left[ \mathcal{L}^S(\mathcal{Q}, \mathcal{D}, \mathcal{S}, \tilde{\mathbf{m}}) \leq \hat{\mathcal{L}}^S(\mathcal{Q}, \mathcal{S}) + \left( \frac{1}{\lambda_1} + \frac{1}{\lambda_2} \right) KL(\mathcal{Q} \| \mathcal{P}) + \frac{1}{\lambda_1} \sum_{i=1}^n \mathop{\mathbb{E}}_{P \sim \mathcal{Q}} KL\left( \mathbb{Q}^*(P, \mathcal{S}_i) \| P \right) \right.$$
$$\left. + \frac{\lambda_1 (b-a)^2}{8n^2} \sum_{i=1}^n \frac{1}{m_i} + \frac{\lambda_2}{8} \sum_{i=1}^n \Delta_i^2 + \ln\left(\frac{1}{\delta}\right) \right] \geq 1 - \delta. \tag{27}$$

**Step 3.** In the final step, we utilize the closed-form expression of the optimal posterior for each client, $\mathbb{Q}^*$, based on Corollary 3.1.1, to simplify (27). By substituting the definition of the server-level empirical loss given in (8) into (27) and refactoring some terms, one obtains:

$$Pr\Big[\mathcal{L}^S(\mathcal{Q},\mathcal{D},\mathcal{S},\tilde{\mathbf{m}}) \leq \underset{P\sim\mathcal{Q}}{\mathbb{E}}\frac{1}{n}\sum_{i=1}^n\big(\hat{\mathcal{L}}^C(\mathbb{Q}^*(P,\mathcal{S}_i),\mathcal{S}_i) + \frac{n}{\lambda_1}KL\big(\mathbb{Q}^*(P,\mathcal{S}_i)\|P\big)\big) \qquad (\ddagger)$$

$$+ \big(\frac{1}{\lambda_1}+\frac{1}{\lambda_2}\big)KL(\mathcal{Q}\|\mathcal{P})$$

$$+ \frac{\lambda_1(b-a)^2}{8n^2}\sum_{i=1}^n\frac{1}{m_i} + \frac{\lambda_2}{8}\sum_{i=1}^n\Delta_i^2 + \ln\big(\frac{1}{\delta}\big)\Big] \geq 1-\delta.$$

When we set $\lambda_1 = n\beta$, each term in the summation in ($\ddagger$) becomes equivalent to the upper bound for client $i$ in Theorem 3.1 up to a constant. This choice of $\lambda_1$ enables us to simplify ($\ddagger$) similar to (6), resulting in:

$$Pr\Big[\mathcal{L}^S(\mathcal{Q},\mathcal{D},\mathcal{S},\tilde{\mathbf{m}}) \leq \frac{-1}{n\beta}\sum_{i=1}^n\underset{P\sim\mathcal{Q}}{\mathbb{E}}\ln Z_\beta^C(P,\mathcal{S}_i) + \big(\frac{1}{n\beta}+\frac{1}{\lambda_2}\big)KL(\mathcal{Q}\|\mathcal{P})$$

$$+ \frac{\beta(b-a)^2}{8n}\sum_{i=1}^n\frac{1}{m_i} + \frac{\lambda_2}{8}\sum_{i=1}^n\Delta_i^2 + \ln\big(\frac{1}{\delta}\big)\Big] \geq 1-\delta.$$

Note that the selection of $\lambda_1 = n\beta$ is only feasible when $\beta \geq 1/n$ because the first step of this proof requires $\lambda_1 \geq 1$. This is the reason behind the assumption $\beta \geq 1/n$ in Theorem 4.1. By substituting $\lambda_2 = \lambda/(n_2 + \upsilon)$, which was motivated in the second step, we get the desired bound. Finally, we use the factorization proposed in Rothfuss et al. (2021) to convert $\ln(1/\delta)$ into $\ln(1/\delta)/\sqrt{n}$.

### 8.2.2 PROOF OF LEMMA 4.2

Assume $\tilde{m}_i$ is only known when $i \in A$, where $A \subset \{1,\cdots,n\}$ is a subset of clients, but is unknown for the rest of the clients, $i \in B = \{1,\cdots,n\}\backslash A$. In an extreme case, $A = \varnothing$ means that $\tilde{m}_i$ is unknown for all clients. Let $\rho(A)$ and $\rho(B)$ denote the number of clients with $\tilde{m}_i > 0$ in sets $A$ and $B$ respectively. It is clear from the definition that $\rho(A) + \rho(B) = n_2$, where $\rho(A)$ is known but $\rho(B)$ and $n_2$ are unknown. Also, $\rho(A) \leq |A|$ and $\rho(B) \leq |B|$, where $|A|$ and $|B|$ stand for the cardinality of the respective set.

We prove that the upper bound in Theorem 4.1 is looser when replacing $\Delta_i$ with $(b-a)/n$ for $i \in B$ and $n_2$ with $\rho(A) + |B|$. It is enough to show:

$$\frac{\rho(A)+\rho(B)+\upsilon}{\lambda}KL(\mathcal{Q}\|\mathcal{P}) + \frac{\sum_{i\in A}\Delta_i^2 + \sum_{i\in B}\Delta_i^2}{8\big(\rho(A)+\rho(B)+\upsilon\big)}\lambda \leq$$

$$\frac{\rho(A)+|B|+\upsilon}{\lambda}KL(\mathcal{Q}\|\mathcal{P}) + \frac{\sum_{i\in A}\Delta_i^2 + |B|(\frac{b-a}{n})^2}{8\big(\rho(A)+|B|+\upsilon\big)}\lambda.$$

Since $\rho(B) \leq |B|$ and KL divergence is non-negative, we will prove that:

$$\frac{\sum_{i\in A}\Delta_i^2 + \sum_{i\in B}\Delta_i^2}{\rho(A)+\rho(B)+\upsilon} \leq \frac{\sum_{i\in A}\Delta_i^2 + |B|(\frac{b-a}{n})^2}{\rho(A)+|B|+\upsilon}. \qquad (28)$$

For clients in $B$ with $\tilde{m}_i = 0$, $\Delta_i = 0$, and for the rest, $\Delta_i \leq (b-a)/n$. Hence, $\sum_{i\in B}\Delta_i^2 \leq \rho(B)\big((b-a)/n\big)^2$ and a stronger condition than (28) is:

$$\frac{\sum_{i\in A}\Delta_i^2 + \rho(B)(\frac{b-a}{n})^2}{\rho(A)+\rho(B)+\upsilon} \leq \frac{\sum_{i\in A}\Delta_i^2 + |B|(\frac{b-a}{n})^2}{\rho(A)+|B|+\upsilon} \iff \sum_{i\in A}\Delta_i^2 \leq \big(\rho(A)+\upsilon\big)\big(\frac{b-a}{n}\big)^2,$$

which holds since $\sum_{i\in A}\Delta_i^2 \leq \rho(A)\big((b-a)/n\big)^2$ and $\upsilon \geq 0$.

### 8.2.3 PROOF OF COROLLARY 4.2.1

The optimal hyper-posterior minimizes the upper bound on the true server-level risk established in 9. By utilizing the structural similarity between the server-level and client-level upper-bounds, equations (9) and (4), we employ Corollary 3.1.1 to obtain $\mathcal{Q}^*$.

### 8.2.4 PROOF OF LEMMA 4.4

We invoke Lemma 8.1 by defining $X_i = (\mathcal{D}_i, m_i, \mathcal{S}_i)$ and using $X_i = (X_i[\mathcal{D}], X_i[m], X_i[\mathcal{S}])$ to distinguish between components in $X_i$. The distribution over $X_i$ is $\mu_i = (\mathcal{T}, X_i[\mathcal{D}]^{X_i[m]})$. Additionally, $l = n$, $f = P$, $\pi = \mathcal{P}$, $\rho = \mathcal{Q}^*$, and $g_i(f, X_i) = \frac{1}{n}\mathcal{L}^C(\mathbb{Q}^*(f, X_i[\mathcal{S}]), X_i[\mathcal{D}])$ are used. While $g_i$ defined above resembles the one from the second step of Proposition 8.2.1, the difference lies in considering $\mathcal{S}_i$ and $\mathcal{D}_i$ as part of the random variable $X_i$, giving rise to their presence in the expectation. By applying Lemma (8.1) with $\gamma = \tilde{\lambda} \geq 1$, one obtains:

$$\mathop{\mathbb{E}}_{(\mathcal{D}_\iota, \tilde{m}_\iota) \sim \mathcal{T}} \mathop{\mathbb{E}}_{\tilde{\mathcal{S}}_\iota \sim \mathcal{D}_\iota^{\tilde{m}_\iota}} \mathop{\mathbb{E}}_{P \sim \mathcal{Q}^*} \mathcal{L}^C(\mathbb{Q}^*(P, \tilde{\mathcal{S}}_\iota), \mathcal{D}_\iota) \leq \frac{1}{n} \mathop{\mathbb{E}}_{P \sim \mathcal{Q}^*} \sum_{i=1}^{n} \mathcal{L}^C(\mathbb{Q}^*(P, \mathcal{S}_i), \mathcal{D}_i)$$
$$+ \frac{1}{\tilde{\lambda}} KL(\mathcal{Q}^* \| \mathcal{P}) + \frac{1}{\tilde{\lambda}} \tilde{\psi}(\tilde{\lambda}). \tag{29}$$

The left-hand side of (29) is the true loss for a generic new client, $\iota$, sampled from $\mathcal{T}$. The moment generating function and the bound on its expectation due to Corollary 8.2.1 are:

$$\tilde{\psi}(\tilde{\lambda}) := \ln \mathop{\mathbb{E}}_{P \sim \mathcal{P}} e^{\tilde{\lambda}\left(\mathbb{E}_{(\mathcal{D}_\iota, \tilde{m}_\iota) \sim \mathcal{T}} \mathbb{E}_{\tilde{\mathcal{S}}_\iota \sim \mathcal{D}_\iota^{\tilde{m}_\iota}} \mathcal{L}^C(\mathbb{Q}^*(P, \tilde{\mathcal{S}}_\iota), \mathcal{D}_\iota) - \frac{1}{n}\sum_{i=1}^{n} \mathcal{L}^C(\mathbb{Q}^*(P, \mathcal{S}_i), \mathcal{D}_i)\right)},$$

$$\mathop{\mathbb{E}}_{(\mathcal{D}_1, m_1) \sim \mathcal{T}} \cdots \mathop{\mathbb{E}}_{(\mathcal{D}_n, m_n) \sim \mathcal{T}} \mathop{\mathbb{E}}_{\mathcal{S}_1 \sim \mathcal{D}_1^{m_1}} \cdots \mathop{\mathbb{E}}_{\mathcal{S}_n \sim \mathcal{D}_n^{m_n}} e^{\frac{1}{\tilde{\lambda}}\tilde{\psi}(\tilde{\lambda})} \leq e^{\frac{\tilde{\lambda}}{8n}(b-a)^2}. \tag{30}$$

Merging (6) and the result of Step 1 in the proof of Proposition 8.2.1 when $\lambda_1 = 1/(n\beta)$ and $\mathbb{Q}^*$ is the optimal posterior mapping in. 5, we rewrite (29) as:

$$\mathop{\mathbb{E}}_{\mathcal{D}_\iota \sim \mathcal{T}} \mathop{\mathbb{E}}_{\tilde{\mathcal{S}}_\iota \sim \mathcal{D}_\iota^{\tilde{m}_\iota}} \mathop{\mathbb{E}}_{P \sim \mathcal{Q}^*} \mathcal{L}^C(\mathbb{Q}^*(P, \tilde{\mathcal{S}}_\iota), \mathcal{D}_\iota) \leq \frac{-1}{n\beta} \sum_{i=1}^{n} \mathop{\mathbb{E}}_{P \sim \mathcal{Q}^*} \ln Z_\beta(P, \mathcal{S}_i)$$
$$+ \left(\frac{1}{\tilde{\lambda}} + \frac{1}{n\beta}\right) KL(\mathcal{Q}^* \| \mathcal{P}) + \frac{1}{\tilde{\lambda}}\tilde{\psi}(\tilde{\lambda}) + \frac{1}{n\beta}\psi_1(n\beta).$$

Plugging in the formula of $\mathcal{Q}^*$ given in Corollary (4.2.1) and setting $\tilde{\lambda} = \lambda/(n_2 + \upsilon)$,

$$\mathop{\mathbb{E}}_{\mathcal{D}_\iota \sim \mathcal{T}} \mathop{\mathbb{E}}_{\tilde{\mathcal{S}}_\iota \sim \mathcal{D}_\iota^{\tilde{m}_\iota}} \mathop{\mathbb{E}}_{P \sim \mathcal{Q}^*} \mathcal{L}^C(\mathbb{Q}^*(P, \tilde{\mathcal{S}}_\iota), \mathcal{D}_\iota) \leq -\left(\frac{1}{n\beta} + \frac{n_2 + \upsilon}{\lambda}\right) \ln Z_\tau^S(\mathcal{P}, \mathcal{S})$$
$$+ \frac{1}{n\beta}\psi_1(n\beta) + \frac{n_2 + \upsilon}{\lambda}\tilde{\psi}\left(\frac{\lambda}{n_2 + \upsilon}\right).$$

Using a similar technique to the proof of Proposition 8.2.1, along with (17) and (30), we obtain the desired result.

### 8.2.5 LOOSER UPPER BOUND FOR NEW CLIENTS

We consider Theorem 4.1 when $\mathcal{Q} = \mathcal{Q}^*$ and simplify the terms in the upper bound that involve $\mathcal{Q}^*$:

$$\frac{-1}{n\beta} \sum_{i=1}^{n} \mathop{\mathbb{E}}_{P \sim \mathcal{Q}^*} \ln Z_\beta^C(P, \mathcal{S}_i) + \left(\frac{1}{n\beta} + \frac{n_2 + \upsilon}{\lambda}\right) KL(\mathcal{Q}^* \| \mathcal{P})$$
$$= \frac{1}{n\beta\tau} \mathop{\mathbb{E}}_{P \sim \mathcal{Q}^*} \left(-\tau \sum_{i=1}^{n} \ln Z_\beta^C(P, \mathcal{S}_i) + \ln \frac{\mathcal{Q}^*(P)}{\mathcal{P}(P)}\right) \tag{31}$$
$$= -\frac{1}{n\beta\tau} \ln Z_\tau^S(\mathcal{P}, \mathcal{S}) = -\left(\frac{1}{n\beta} + \frac{n_2 + \upsilon}{\lambda}\right) \ln Z_\tau^S(\mathcal{P}, \mathcal{S}). \tag{32}$$

Equation (31) follows from the KL divergence definition and (32) is obtained by substituting the closed-form formula of $\mathcal{Q}^*$ derived in Corollary 4.2.1 into the expression. To compare the bound in Theorem 4.1 with Lemma 4.4, we can substitute the first two terms in Theorem 4.1 with (32). In order for the bound in Lemma 4.4 to be looser than this substituted bound, it is sufficient to show that $\sum_{i=1}^{n} \Delta_i^2 \leq (b-a)^2/n$, which is true based on the definition of $\Delta_i$ in Theorem 4.1.

### 8.3 DETAILS OF THE ALGORITHM

#### 8.3.1 ASYMPTOTIC BEHAVIOR AND NON-VACUOUSNESS OF BOUNDS

We analyze the asymptotic behavior of the client-level and server-level bounds as the number of samples per client and the number of existing clients approach infinity, i.e., $m_i \to \infty$ and $n \to \infty$. A PAC bound is considered consistent when the gap between the true and empirical risks goes to zero. The client-level and server-level bounds are consistent if and only if: a) $\beta \in \Omega(1)$, b) $\beta \in o(m_i)$, c) $\lambda \in \Omega(n_2)$, d) $\lambda \in o(n(n_2 + v))$. Here, $o$ and $\Omega$ represent the small-oh and big-omega notations for function growth rate (Cormen et al., 2022).

Non-vacuous bounds are essential, ensuring that the terms independent of the posterior or the hyper-posterior are not excessively large such that the bound on the true loss holds regardless of the empirical loss. To achieve a non-vacuous bound, it is necessary (but not sufficient) that $b - a < 8$ and that $\epsilon_i < \sqrt{2(b-a)}$. The latter condition can be converted into an upper bound on $\lambda$ based on the results from Lemma 4.3.

In our experiments, we follow Rothfuss et al. (2021) and set $\beta = m_i + \tilde{m}_i$, leading to a non-vanishing gap between the true and empirical risks at the client and server levels. However, this choice simplifies the computations as discussed in Section 5. Moreover, this choice leads to faster decay of the KL term in both the client-level and server-level bounds, which can be advantageous when $m_i$ is small (Rothfuss et al., 2021). We tune $\lambda$, while respecting the non-vacuousness condition, to manipulate the regularization strength of the hyper-posterior towards the hyper-prior.

#### 8.3.2 BACKGROUND ON GP

Below, we provide additional details regarding the GP models we use. For a more comprehensive overview on GPs, please refer to Rasmussen and Williams (2005). In the rest of this section, we express the dataset of client $i$ as $\mathcal{S}_i = (\mathbf{X}_i, \mathbf{y}_i)$, where $\mathbf{X}_i \in R_{\mathbf{x}}^{m_i} \subset \mathbb{R}^{m_i \times d}$ is a matrix with row $j$ representing the features of the $j$-th sample in $\mathcal{S}_i$ denoted as $\mathbf{x}_{ij} \in R_{\mathbf{x}}$. The corresponding target values are stored in the vector $\mathbf{y}_i \in R_y^{m_i} \subset \mathbb{R}^{m_i}$, with row $j$ representing the target value $y_{ij} \in R_y$.

**GP with a deep mean and a deep kernel (Wilson et al., 2016).** Let $P_{\boldsymbol{\phi}}(h) = \mathcal{GP}(h|m_{\boldsymbol{\phi}}(\mathbf{x}), k_{\boldsymbol{\phi}}(\mathbf{x}, \mathbf{x}'))$ denote a GP prior specified by a deep mean function, $m_{\boldsymbol{\phi}}$, a deep kernel function, $k_{\boldsymbol{\phi}}$, and a Gaussian likelihood with noise standard deviation, $\sigma_{\boldsymbol{\phi}} \in \mathbb{R}_+$. The vector $\boldsymbol{\phi} \in \mathbb{R}^{d_{\boldsymbol{\phi}}}$ concatenates all learnable hyper-parameters of the GP prior, including the mean parameters, kernel parameters, and the likelihood noise. The mean function, $m_{\boldsymbol{\phi}} : R_{\mathbf{x}} \to \mathbb{R}$, is implemented as a multi-layer NN with weights given by $\boldsymbol{\phi}$, hyperbolic tangent activation functions in the hidden layers, and linear output functions in the output layer. The kernel function is a squared-exponential (SE) kernel applied on top of an NN, defined as:

$$k_{\boldsymbol{\phi}}(\mathbf{x}, \mathbf{x}') := exp(\frac{-1}{2}\|f_{\boldsymbol{\phi}}(\mathbf{x}) - f_{\boldsymbol{\phi}}(\mathbf{x}')\|_2^2) \in [0, 1].$$

The function $f_{\boldsymbol{\phi}} : R_{\mathbf{x}} \to \mathbb{R}^{d_f}$ represents an NN with weights given by $\boldsymbol{\phi}$ that maps the typically high-dimensional feature vector, $\mathbf{x} \in R_{\mathbf{x}} \subset \mathbb{R}^d$, to a lower-dimensional output vector in $\mathbb{R}^{d_f}$, where in our specific case, $d_f$ is two. Deep kernels serve as feature extractors and allow for learning more sophisticated representations from the data (Ober et al., 2021). The length scale of the SE kernel is set to 1 because the weights of the output layer of $f_{\boldsymbol{\phi}}$ can be freely chosen to compensate for it.

**Computing the LML.** As mentioned in Section 5, when employing a GP prior, negative log likelihood loss, and setting $\beta = m_i$, the quantity $\ln Z_{m_i}^C(P, \mathcal{S}_i)$ defined in Corollary 3.1.1 corresponds to the LML of the GP. For client $i$ using the prior $P_{\boldsymbol{\phi}_\kappa}$, the closed-form formula for the LML is as follows:

$$\ln Z_{m_i}^C(P, \mathcal{S}_i) = \ln \Pr[\mathbf{y}_i|\mathbf{X}_i] = -\frac{1}{2}(\mathbf{y}_i - \mathbf{m}_{\kappa,i})^\top (\mathbf{K}_{\kappa,i} + \sigma_{\boldsymbol{\phi}_\kappa}^2 \mathbf{I})^{-1}(\mathbf{y}_i - \mathbf{m}_{\kappa,i})$$

$$- \frac{1}{2}\ln|\mathbf{K}_{\kappa,i} + \sigma_{\boldsymbol{\phi}_\kappa}^2 \mathbf{I}| - \frac{m_i}{2}\ln(2\pi), \tag{33}$$

where $\mathbf{m}_{\kappa,i} \in \mathbb{R}^{m_i}$ is given by $[m_{\boldsymbol{\phi}_\kappa}(\mathbf{x}_{i1}), \cdots, m_{\boldsymbol{\phi}_\kappa}(\mathbf{x}_{im_i})]^T$ and $|\mathbf{A}|$ denotes the determinant of the matrix $\mathbf{A}$. The matrix $\mathbf{K}_{\kappa,i} \in \mathbb{R}^{m_i \times m_i}$ is the kernel matrix associated with the kernel function

$k_{\phi_\kappa}$ applied to the feature matrix $\mathbf{X}_i$. Furthermore, $\mathbf{K}_{\kappa,i,*} \in \mathbb{R}^{m_i}$ is computed using the kernel function $k_{\phi_\kappa}$ between the feature matrix $\mathbf{X}_i$ and the test features $\mathbf{x}_*$.

The determinant term is commonly viewed as a complexity penalty for the kernel (Rasmussen and Williams, 2005). However, recent findings in Rothfuss et al. (2021) suggest that this form of complexity regularization may be inadequate when dealing with expressive kernels that possess numerous hyperparameters, such as deep kernels. Furthermore, there is no complexity penalty imposed on the prior mean, which can lead to a higher risk of overfitting. In PAC-PFL, we address these limitations by using (9) as the loss function, which includes a KL divergence term between the hyper-posterior and the hyper-prior. The KL term penalizes hyper-posteriors that deviate significantly from the hyper-prior, thereby effectively regularizing both the mean and the kernel of the GP prior.

**Computing the predictive posterior.** Given a set of priors $P_{\phi_1}, \cdots, P_{\phi_k}$ trained by PAC-PFL, our objective is to compute the predictive posterior at a test point $\mathbf{x}_*$. We denote by $\hat{\mathcal{Q}}$ a uniform distribution over these priors, which serves as the SVGD approximation to the true optimal hyper-posterior, $\mathcal{Q}^*$. By introducing a categorical random variable $Z$ that is distributed uniformly over $1, \cdots, k$ and denotes which of the priors $P_{\phi_1}, \cdots, P_{\phi_k}$ is used for making inference, the predictive posterior can be expressed as follows:

$$
\begin{aligned}
\Pr[y_*|\mathbf{x}_*, \mathcal{S}_i, \hat{\mathcal{Q}}] &= \sum_{\kappa=1}^{k} \Pr[y_*|\mathbf{x}_*, \mathcal{S}_i, \hat{\mathcal{Q}}, Z=\kappa] \Pr[Z=\kappa|\mathbf{x}_*, \mathcal{S}_i, \hat{\mathcal{Q}}] \\
&= \sum_{\kappa=1}^{k} \Pr[y_*|\mathbf{x}_*, \mathcal{S}_i, P_{\phi_\kappa}] \Pr[Z=\kappa|\mathbf{x}_*, \mathcal{S}_i, \hat{\mathcal{Q}}] \\
&= \sum_{\kappa=1}^{k} \Pr[y_*|\mathbf{x}_*, \mathcal{S}_i, P_{\phi_\kappa}] \Pr[\mathbf{x}_*, \mathcal{S}_i|P_{\phi_\kappa}] \frac{\Pr[Z=k|\hat{\mathcal{Q}}]}{\Pr[\mathbf{x}_*, \mathcal{S}_i|\hat{\mathcal{Q}}]} \quad (\textit{Bayes rule}) \\
&= \frac{1/k}{\Pr[\mathbf{x}_*, \mathcal{S}_i|\hat{\mathcal{Q}}]} \sum_{\kappa=1}^{k} \Pr[y_*|\mathbf{x}_*, \mathcal{S}_i, P_{\phi_\kappa}] \Pr[\mathbf{x}_*, \mathcal{S}_i|P_{\phi_\kappa}] \quad (\hat{\mathcal{Q}} \textit{ is uniform}). \quad (34)
\end{aligned}
$$

The first term in (34) is the predictive posterior distribution for client $i$ corresponding to the GP prior $P_{\phi_\kappa}(h) = \mathcal{GP}(h|m_{\phi_\kappa}, k_{\phi_\kappa})$, which is given by Rasmussen and Williams (2005) as:

$$
\Pr[y_*|\mathbf{x}_*, \mathcal{S}_i, P_{\phi_\kappa}] = \mathcal{N}(y_*; \mu_\kappa, \Sigma_\kappa), \tag{35a}
$$

$$
\mu_\kappa := m_{\phi_\kappa}(\mathbf{x}_*) + \mathbf{K}_{\kappa,i,*}^T (\mathbf{K}_{\kappa,i} + \sigma_{\phi_\kappa}^2 \mathbf{I})^{-1} (\mathbf{y}_i - \mathbf{m}_{\kappa,i}), \tag{35b}
$$

$$
\Sigma_\kappa := k_{\phi_\kappa}(\mathbf{x}_*, \mathbf{x}_*) - \mathbf{K}_{\kappa,i,*}^T (\mathbf{K}_{\kappa,i} + \sigma_{\phi_\kappa}^2 \mathbf{I})^{-1} \mathbf{K}_{\kappa,i,*} + \sigma_{\phi_\kappa}^2 \mathbf{I}. \tag{35c}
$$

Based on the assumption that $\mathbf{x}_*$ is independent of all samples in $\mathcal{S}_i$, we can expand the second term in (34) as follows:

$$
\begin{aligned}
\Pr[\mathbf{x}_*, \mathcal{S}_i|P_{\phi_\kappa}] &= \Pr[\mathbf{x}_*|P_{\phi_\kappa}] \Pr[\mathbf{X}_i|P_{\phi_\kappa}] \Pr[\mathbf{y}_i|\mathbf{X}_i, P_{\phi_\kappa}] \\
&= \Pr[\mathbf{x}_*] \Pr[\mathbf{X}_i] \Pr[\mathbf{y}_i|\mathbf{X}_i, P_{\phi_\kappa}] \\
&= \Pr[\mathbf{x}_*] \Pr[\mathbf{X}_i] \mathcal{N}(\mathbf{y}_i; m_{\phi_\kappa}(\mathbf{X}_i), k_{\phi_\kappa}(\mathbf{X}_i, \mathbf{X}_i) + \sigma_{\phi_\kappa}^2 \mathbf{I}), \quad (36)
\end{aligned}
$$

where the last line is the predictive distribution of a GP before conditioning on the observed data.

By merging (34)-(36), one obtains

$$
\Pr[y_*|\mathbf{x}_*, \mathcal{S}_i, \hat{\mathcal{Q}}] = \sum_{\kappa=1}^{k} \alpha_\kappa \mathcal{N}(y_*; \mu_\kappa, \Sigma_\kappa), \tag{37a}
$$

$$
\alpha_\kappa := \frac{\Pr[\mathbf{x}_*] \Pr[\mathbf{X}_i]}{k \Pr[\mathbf{x}_*, \mathcal{S}_i|\hat{\mathcal{Q}}]} \mathcal{N}(\mathbf{y}_i; m_{\phi_\kappa}(\mathbf{X}_i), k_{\phi_\kappa}(\mathbf{X}_i, \mathbf{X}_i) + \sigma_{\phi_\kappa}^2 \mathbf{I}) \in [0, 1] \tag{37b}
$$

It is straightforward to show that $\sum_{\kappa=1}^{k} \alpha_\kappa = 1$ in equation (37b). As a result, the predictive posterior in equation (37a) is a valid distribution over $y_*$.

### 8.3.3 BACKGROUND ON BNN

In this section, we explore using Bayesian Neural Networks (BNNs) as a classification tool. Let $h_{\boldsymbol{\theta}} : R_{\mathbf{x}} \to R_y$ represent a neural network, with $\boldsymbol{\theta}$ denoting its weight parameters. Utilizing this mapping, we establish the conditional distribution as a Categorical distribution, derived as follows: $\Pr[y|\boldsymbol{x}, \boldsymbol{\theta}] = Categorical\big(softmax\big(h_{\boldsymbol{\theta}}(\boldsymbol{x})\big)\big)$. Unlike for GPs, the LML,

$$\ln Z_{\beta}(\mathcal{S}_i, P_{\boldsymbol{\phi}}) = \ln \mathbb{E}_{\boldsymbol{\theta} \sim P_{\boldsymbol{\phi}}} \big[ e^{-\beta \hat{\mathcal{L}}^C \big( \mathbb{Q}(P_{\boldsymbol{\theta}}, \mathcal{S}_i \cup \tilde{\mathcal{S}}_i), \mathcal{S}_i \cup \tilde{\mathcal{S}}_i \big)} \big],$$

is intractable for BNNs. Instead, we use the following formula, as proposed by Rothfuss et al. (2021), to approximate the LML:

$$\ln \tilde{Z}_{\beta}(\mathcal{S}_i, P_{\boldsymbol{\phi}}) = LSE_{j=1}^{L} \Big( -\beta \hat{\mathcal{L}}^C \big( \mathbb{Q}(P_{\boldsymbol{\theta}_j}, \mathcal{S}_i \cup \tilde{\mathcal{S}}_i), \mathcal{S}_i \cup \tilde{\mathcal{S}}_i \big) \Big) - \ln L,$$

where *LSE* is the LogSumExp function. This formula involves drawing $L$ samples $\theta_1, \cdots, \theta_L$ from $P_{\boldsymbol{\phi}}$ to approximate the LML. We utilize this formula for LML approximation in the context of BNNs.

### 8.3.4 BACKGROUND ON SVGD

SVGD approximates a target probability distribution using a discrete uniform distribution over a set of designated samples called *particles* (Liu and Wang, 2016). Through an iterative process, the particles are updated by minimizing the KL divergence between the estimated and the target distributions in the reproducing kernel Hilbert space corresponding to a kernel function, $k_{SVGD}$. We utilize an RBF (Radial Basis Function) kernel with a heuristically chosen length scale, as described in Liu and Wang (2016). In our framework, we aim to obtain samples $P_{\boldsymbol{\phi}_1}, \cdots, P_{\boldsymbol{\phi}_k}$ from the target distribution $\mathcal{Q}^*$. To simplify the notation, we represent the particles as $\boldsymbol{\phi}_1, \cdots, \boldsymbol{\phi}_k$, where each particle $\boldsymbol{\phi}_{\kappa}$ fully characterizes the corresponding prior, $P_{\boldsymbol{\phi}_{\kappa}}$. Accordingly, we write $\mathcal{P}(\boldsymbol{\phi}_{\kappa})$ and $\mathcal{Q}^*(\boldsymbol{\phi}_{\kappa})$ instead of $\mathcal{P}(P_{\boldsymbol{\phi}_{\kappa}})$ and $\mathcal{Q}^*(P_{\boldsymbol{\phi}_{\kappa}})$.

Initially, the particles are sampled independently from the hyper-prior. Then, at each iteration, each particle $\boldsymbol{\phi}_{\kappa}$ for $\kappa \in \{1, \cdots, k\}$ is updated according to the following rule:

$$\boldsymbol{\phi}_{\kappa} \leftarrow \boldsymbol{\phi}_{\kappa} + \frac{\eta}{k} \sum_{l=1}^{k} \big( k_{SVGD}(\boldsymbol{\phi}_l, \boldsymbol{\phi}_{\kappa}) \nabla_{\boldsymbol{\phi}_l} \ln \mathcal{Q}^*(\boldsymbol{\phi}_l) + \nabla_{\boldsymbol{\phi}_l} k_{SVGD}(\boldsymbol{\phi}_l, \boldsymbol{\phi}_{\kappa}) \big), \quad (38)$$

where $\eta$ is the learning rate at the current iteration. By utilizing the formula for $\mathcal{Q}^*$ provided in Corollary 4.2.1 when $\beta = m_i$, we can derive the following expression for $\kappa \in \{1, \cdots, k\}$:

$$\nabla_{\boldsymbol{\phi}_{\kappa}} \ln \mathcal{Q}^*(\boldsymbol{\phi}_{\kappa}) = \nabla_{\boldsymbol{\phi}_{\kappa}} \ln \mathcal{P}(\boldsymbol{\phi}_{\kappa}) + \tau \sum_{i=1}^{n} \nabla_{\boldsymbol{\phi}_{\kappa}} \ln Z_{m_i}^C(P_{\boldsymbol{\phi}_{\kappa}}, \mathcal{S}_i), \quad (39)$$

where $\tau$ is defined in Corollary 4.2.1. By comparing (38) and (39), it becomes evident that the data of client $i$ is solely involved in the SVGD update through the term $\nabla_{\boldsymbol{\phi}_{\kappa}} \ln Z_{m_i}^C(P_{\boldsymbol{\phi}_{\kappa}}, \mathcal{S}_i)$ for each particle $\boldsymbol{\phi}_{\kappa}$. As a result, the clients only need to transmit the gradient vector, $[\nabla_{\boldsymbol{\phi}_1} \ln Z_{m_i}^C(P_{\boldsymbol{\phi}_1}, \mathcal{S}_i), \cdots, \nabla_{\boldsymbol{\phi}_k} \ln Z_{m_i}^C(P_{\boldsymbol{\phi}_k}, \mathcal{S}_i)]^T$, or an approximation of it using a mini-batch approach, to the server. This observation is the intuition behind the sub-routine *Client_Update* in Algorithm 1 which we present in detail in the next subsection.

### 8.3.5 ALGORITHM SUB-ROUTINES

In light of the GP formulas and the SVGD tool outlined in Appendices 8.3.2 and 8.3.4, we proceed to describe the *Client_Update* and *SVGD_Update* sub-routines of Algorithm 1. As stated in Appendix 8.3.4, client $i$ is required to transmit $[\nabla_{\boldsymbol{\phi}_{\kappa}} \ln Z_{m_i}^C(P_{\boldsymbol{\phi}_{\kappa}}, \mathcal{S}_i)]$ to the server for every $\kappa \in 1, \cdots, k$. However, calculating $\ln Z_{m_i}^C(P_{\boldsymbol{\phi}_{\kappa}}, \mathcal{S}_i)]$ as derived in (33) involves inverting an $m_i \times m_i$ matrix, which can be computationally demanding for clients with large datasets. Therefore, we use an approximation $\ln Z_{m_i}^C(P_{\boldsymbol{\phi}_{\kappa}}, \mathcal{S}_i^{(b)})$ computed on a mini-batch $\mathcal{S}_i^{(b)}$ extracted from the data of client $i$. A pseudocode of the described procedure for client $i$ is presented in Algorithm 2.

---

**Algorithm 2** Client_Update for client $i$ with dataset $\mathcal{S}_i$. Requires: mini-batch size ($b$), current particles ($\phi_1, \cdots, \phi_k$).

---

1: **Client $i$ executes:**
2:      Sample mini-batch $\mathcal{S}_i^{(b)}$ of size $b$ from $\mathcal{S}_i$
3:      **for** $\kappa = 1$ to $k$ **do**
4:          Compute $\nabla_{\phi_\kappa} \ln Z_{m_i}^C(P_{\phi_\kappa}, \mathcal{S}_i^{(b)})$ by performing automatic differentiation over (33)
5:      $\mathbf{G}_i \leftarrow [\nabla_{\phi_1} \ln Z_{m_i}^C(P_{\phi_1}, \mathcal{S}_i^{(b)}), \cdots, \nabla_{\phi_k} \ln Z_{m_i}^C(P_{\phi_k}, \mathcal{S}_i^{(b)})]^T$
6:      **return** $\mathbf{G}_i$

---

The procedure takes as input the particles at the current iteration, $\phi_1, \cdots, \phi_k$, and the mini-batch size, $b$. It computes the gradients $\nabla_{\phi_\kappa} \ln Z_{m_i}^C(P_{\phi_\kappa}, \mathcal{S}_i^{(b)})$ for each particle $\phi_\kappa$ over a mini-batch $\mathcal{S}_i^{(b)}$. Finally, the gradients are returned as the output of the procedure, $\mathbf{G}_i \in \mathbb{R}^{k \times d_\phi}$.

After collecting and aggregating client updates from the subset of sampled clients $\mathcal{C}_t$ in Line 7 of Algorithm 1, the SVGD update is executed as per Algorithm 3.

---

**Algorithm 3** SVGD_Update. Requires: learning rate ($\eta$), parameter ($\tau$), aggregated gradients ($\mathbf{G}$), hyper-prior ($\mathcal{P}$), current particles ($\phi_1, \cdots, \phi_k$).

---

1: **Server executes:**
2:      **for** $\kappa = 1$ to $k$ **do**
3:          $\nabla_{\phi_\kappa} \ln \mathcal{Q}^*(\phi_\kappa) \leftarrow \nabla_{\phi_\kappa} \ln \mathcal{P}(\phi_\kappa) + \tau (G_{\kappa *})^T$
4:      **for** $\kappa = 1$ to $k$ **do**
5:          $\phi_\kappa \leftarrow \phi_\kappa + \frac{\eta}{k} \sum_{l=1}^k \left( k_{SVGD}(\phi_l, \phi_\kappa) \nabla_{\phi_l} \ln \mathcal{Q}^*(\phi_l) + \nabla_{\phi_l} k_{SVGD}(\phi_l, \phi_\kappa) \right)$
6:      **return** $\phi_1, \cdots, \phi_k$

---

In Line 3, $G_{\kappa *}$ refers to the $\kappa$-th row of the matrix $\mathbf{G}$. The server executes Algorithm 3 to update the particles based on the aggregated gradients $\mathbf{G}$ and the learning rate $\eta$. Note that besides the algorithm inputs, $\eta, \mathbf{G}, \phi_1, \cdots, \phi_k$, the procedure also depends on the SVGD kernel, $k_{SVGD}$.

Lastly, it is worth mentioning that if necessary, the mini-batch size, $b$, can vary from client to client. In such cases, the aggregation process in Line 7 of Algorithm 1 becomes a weighted average instead.

### 8.3.6 TABLE OF PARAMETERS

We provide a comprehensive overview of the parameters relevant to our theoretical results and to our algorithm, along with their interrelationships in Table 1. To enhance clarity, the parameters are categorized into three groups, separated by horizontal lines. The first group describes the fundamental attributes of the FL problem, such as the number of clients, and is set externally. In the second group, we list the parameters central to our theoretical findings, elucidating their relations to other parameters. Lastly, the third group encapsulates the parameters used in implementing Algorithm 1, explaining their role in the practical application of our proposed methodology. Importantly, it should be noted that the constraints on the parameters have been deliberately set to ensure non-vacuous bounds, as discussed in Appendix 8.3.1.

### 8.3.7 COMPUTATIONAL COMPLEXITY

In this section, we discuss the computational complexity of a single iteration in training PAC-PFL, specifically when employing the log-likelihood loss and $\beta = m_i$, as outlined in the paper. At the client level, the computational complexity hinges on the calculation of the LML. For GPs, this complexity is $\mathcal{O}(km_i^3)$, where $k$ is the number of SVGD particles, and $m_i$ denotes the size of the training dataset for the specific client $i$. For BNNs, the complexity is $\mathcal{O}(kLm_i)$, where $L$ corresponds to the number of samples used to approximate the LML, as explained in Appendix 8.3.3.

At the server level, the computational complexity varies based on the type of hyper-prior used. When utilizing a hyper-prior with a diagonal covariance matrix, the complexity is $\mathcal{O}(ck + k^2)$, where $c$ represents the number of clients selected per iteration. If a hyper-prior with a full covariance matrix is used, the complexity increases to $\mathcal{O}(ck + k^3)$.

| Param | Description | Constraint | Selection |
|---|---|---|---|
| $n$ | number of clients | $\in \mathbb{N}$ | enforced by the problem |
| $n_2$ | number of clients with new samples | $\in \{0, \cdots, n\}$ | enforced by the problem |
| $m_i$ | number of samples for client $i$ | $\in \mathbb{N}$ | enforced by the problem |
| $\tilde{m}_i$ | number of new samples for client $i$ | $\in \{0, \cdots, m_i\}$ | enforced by the problem |
| $\mathcal{P}$ | hyper-prior | - | $\sigma_{\mathcal{P}}^2 \mathbf{I}$ |
| $\sigma_{\mathcal{P}}^2$ | variance of $\mathcal{P}$ | $\in \mathbb{R}_+$ | tuned by cross-validation |
| $\lambda$ | | $\geq n_2 + \upsilon$ s.t. $\epsilon_i \leq \sqrt{2(b-a)}$ | tuned by CV |
| $\delta$ | confidence level | $\in (0, 1]$ | decided individually by the clients or the server |
| $\upsilon$ | constant in Theorem 4.1 | $\in \mathbb{R}_+$ | $10^{-4}$ |
| $\ell$ | loss | $\in [a, b]$ s.t. $b - a < 8$ | negative log-likelihood |
| $\beta$ | temperature of $Q_i^*$ | $\in \mathbb{R}_+$ | $m_i$ (see Appendix 8.3.1) |
| $\tau$ | temperature of $\mathcal{Q}^*$ | $\in \mathbb{R}_+$ | $\frac{\lambda}{\lambda + \beta n(n_2 + \upsilon)}$ (Corollary 4.2.1) |
| $\epsilon_i$ | DP parameter | $\in \mathbb{R}_+$ | $2\beta\tau(b-a)/m_i$ (Lemma 4.3) |
| $\eta$ | learning rate | $\in \mathbb{R}_+$ | tuned by CV |
| $k$ | number of SVGD particles | $\in \mathbb{N}$ | see code |
| $T$ | number of iterations | $\in \mathbb{N}$ | see code |
| $c$ | number of clients per iteration | $\in \{1, \cdots, n\}$ | see code |
| $b$ | data batch size for each client | $\in \{1, \cdots, m_i\}$ | see code |
| $L$ | number of samples for estimating the MLL of BNNs | $\in \mathbb{N}$ | see code |

Table 1: Parameters involved in our theoretical results and in our algorithm and their interrelations. The constraints are set such that the bounds are non-vacuous. The parameters are categorized into three groups separated by horizontal lines: the first group describes the characteristics of the FL problem. The second group enumerates the parameters used in our theoretical results, while the final group encompasses the parameters involved in Algorithm 1.

## 8.4 ROLE OF DP

We adopt DP in two key ways. Firstly, we utilize DP to ensure that when the server samples a prior distribution from the hyper-posterior and releases it, this published prior distribution does not raise privacy concerns for the existing clients. This application is akin to typical scenarios in FL research where DP is used to protect the model shared by the server from revealing information about clients' data. Besides addressing privacy concerns, we use DP to establish a PAC bound at the client level. As illustrated in Fig. 1, our setup involves prior distributions that depend on the data of existing clients. While most PAC bounds assume the prior distribution was selected before observing any data, we leverage results from Dziugaite and Roy (2018) to derive a PAC bound that holds when a data-dependent prior is obtained through a differentially private algorithm. To the best of our knowledge, DP has not been employed for this purpose in previous FL methods. Below, we discuss the role of DP in our ideal setup (Section 4) and our practical algorithm (Section 5).

**Inherent privacy of the optimal hyper-posterior** In Section 4, we considered a specific family of hyper-posteriors, where sampling a prior from the hyper-posterior satisfies $\epsilon$-DP for a finite $\epsilon$. This assumption simplifies computations and enables us to use the closed-form posterior provided in Corollary 3.1.1. We then established the server-level upper bound in Theorem 4.1 for hyper-posteriors meeting the privacy criterion. Interestingly, the resulting upper bound does not contain the parameter $\epsilon$. This can be intuitively explained by the fact that the hyper-prior is chosen independently of the data, making the server-level scheme akin to typical PAC-Bayesian bounds that employ data-independent priors. Additional details can be found in the proof presented in Appendix 8.2. In summary, the $\epsilon$-DP

assumption facilitates the computations by allowing us to use the optimal posterior formula without directly impacting the server-level bound.

By minimizing the server-level upper bound, we derived the closed-form formula for the optimal hyper-posterior in Corollary 4.2.1. Subsequently, we ensured that the optimal hyper-posterior satisfies the privacy assumption we started with. To achieve this, Lemma 4.3 determined the value of $\epsilon$ for which sampling the prior from the optimal hyper-posterior meets $\epsilon$-DP. As this $\epsilon$ is finite, it confirms that our optimal hyper-posterior belongs to the family of hyper-posteriors we initially considered, thus completing the derivations. While assuming a finite $\epsilon$ is sufficient for deriving the intended results, it is worth noting that the client-level bound becomes looser for larger values of $\epsilon$. Further discussion on providing non-vacuous bounds can be found in Appendix 8.3.1.

In the ideal setup, DP arises from the inherent randomness in sampling from the optimal hyper-posterior. As a result, DP is achieved without the need for externally injecting noise, which is a common practice in typical DP mechanisms. We refer to this property as the *inherent DP* of the optimal hyper-posterior.

**Loss of inherent DP due to SVGD**    As discussed in Section 5, since SVGD is a deterministic sampling algorithm, we forfeit the inherent privacy of $\mathcal{Q}^*$. Consequently, the client-level and server-level bounds no longer hold for the approximate hyper-posterior. Instead, we rely on the premise that if SVGD effectively approximates the optimal hyper-posterior, the empirical and true risks of the approximated and optimal hyper-posteriors should be closely aligned, suggesting the bound's validity.

**Differentially private PAC-PFL**    To reintroduce $\epsilon$-DP after SVGD approximation, Algorithm 1 can be modified, drawing inspiration from differentially private FedAvg (Geyer et al., 2017). In this modified version, the server clips the norm of the gradient sent by each client and introduces noise to the aggregated gradient. For simplicity, we will illustrate the algorithm using only one SVGD particle. A pseudo-code for this private version of PAC-PFL is provided in Algorithm 4.

---

**Algorithm 4** Differentially private PAC-PFL with 1 SVGD particle. Requires: privacy parameter ($\epsilon$), gradient clipping norm ($\gamma$), hyper-prior ($\mathcal{P}$), parameter ($\tau$), number of iterations ($T$), number of clients per iteration ($c$), mini-batch size ($b$), learning rate ($\eta$).

---

1: **Server executes:**
2:     Initialize prior $P_\phi \sim \mathcal{P}$
3:     **for** $t = 1$ to $T$ **do**
4:         Select a random subset $\mathcal{C}_t$ of $c$ clients
5:         **for** each selected client $i$ in $\mathcal{C}_t$ **in parallel do**
6:             $g_i \leftarrow Client\_Update(b, \phi)$
7:             Clip gradient norm: $\hat{g}_i \leftarrow g_i / \max\left(1, \frac{\|g_i\|_2}{\gamma}\right)$
8:             Sample noise: $\nu \sim Lap(\mathbf{0}, \frac{T\gamma}{\epsilon c} \mathbf{I}_{d_\phi})$
9:             Aggregate gradients and inject noise: $\hat{g} \leftarrow \frac{1}{c} \sum_{i \in \mathcal{C}_t} \hat{g}_i + \nu$
10:            Update priors: $\phi \leftarrow SVGD\_Update(\eta, \tau, \hat{g}, \mathcal{P}, \phi)$
11:    **return** differentially private SVGD approximation of $\mathcal{Q}^*$: $P_{\phi_1}$

---

As we are considering a single particle in Algorithm 4, the *Client_Update* function returns a vector $g_i$ instead of a matrix $G_i$, as is the case when using multiple SVGD particles. In Algorithm 4, the notation $Lap(\mathbf{0}, \frac{T\gamma}{\epsilon c} \mathbf{I} d\phi)$ represents a multivariate Laplace probability distribution. In this context, $\mathbf{0}$ denotes a zero mean vector, and $\frac{T\gamma}{\epsilon c} \mathbf{I} d\phi$ specifies the scale parameter applied across all dimensions. Both the particle, $\phi$, and the noise vector, $\nu$, have dimension $d_\phi$.

The noise scale in Algorithm 4 grows linearly with the number of iterations, $T$. This can introduce significant noise into the gradients, potentially impairing both convergence and the overall accuracy of the algorithm. This challenge is intrinsic to differentially private gradient descent and is not unique to our model or FL in general (Bagdasaryan et al., 2019). Investigating alternative strategies to achieve $\epsilon$-DP remains a promising avenue for future research.

We assess the performance of Algorithm 4 using a synthetic toy dataset, introduced in Appendix 8.5. Specifically, we use 120 existing clients, each with 10 samples. We vary the privacy parameter $\epsilon$ and examine its impact on the algorithm's performance, measured by the average RSMSE metric. Figure 3 illustrates the RSMSE across different values of $\epsilon$. Additionally, we plot the RSMSE of the non-private algorithm, Algorithm 1, in red for reference. As anticipated, the model's performance improves as $\epsilon$ increases, signifying a lower level of privacy and consequently reduced noise requirements.

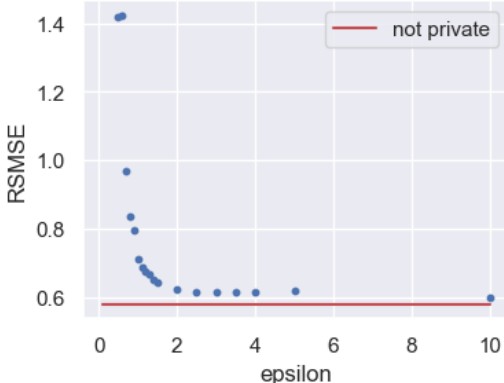

Figure 3: Performance evaluation of differentially private PAC-PFL (Algorithm 4) on the synthetic toy dataset from Appendix 8.5. The RSMSE metric is plotted against varying values for the differential privacy parameter, $\epsilon$. Lower $\epsilon$ values correspond to lower privacy levels. The RSMSE of non-private PAC-PFL (Algorithm 1) is shown in red for reference. As expected, performance improves as the privacy level decreases due to lower noise injection.

## 8.5 DATASETS

### 8.5.1 PV DATASET

The transition to renewable energy sources, such as roof-top photovoltaic (PV) panels, is crucial to address energy challenges; however, the intermittency of solar energy remains a challenge that requires accurate prediction of solar panel power output. Predicting the time series of PV panel power outputs requires either specific measurements or large amounts of data. A potential solution is to design a collaborative methodology among multiple PV datasets collected at nearby locations for predictive modeling.

We work with an hourly simulated dataset of PV generation time series from rooftop panels in Lausanne, Switzerland. Our objective is to predict the next-hour PV generation, utilizing 15 features encompassing auto-regressors and weather data. We investigate four scenarios: *PV-S (150)*, *PV-S (610)*, *PV-EW (150)*, and *PV-EW (610)*. In the *PV-S* variants, all houses face south, while the *PV-EW* variants involve houses oriented either east or west. The *PV-EW* scenarios, being bimodal and highly heterogeneous, present greater modeling challenges. To assess the impact of dataset sizes (denoted as **m**), we consider two settings: clients with either 150 or 610 training samples, corresponding to two-week and two-month data windows, respectively. The number in each dataset's name indicates the training sample size.

**Data generation.** To generate simulated datasets for the PV-S and PV-EW experiments, we first obtained a real dataset of hourly solar radiation and meteorological measurements in Lausanne from the *Photovoltaic Geographical Information System* (PVGIS) online tool (PVG). We then used the *pvlib* Python library (Holmgren et al., 2022) to simulate the PV power output based on these measurements. To simulate the PV panels in different houses, we sampled meteorological data and installation specifications from normal distributions with specified means and standard deviations. For example, in the PV-EW experiment, the azimuths of the PV panels were sampled from a bimodal normal distribution of $0.5\big(\mathcal{N}(90°, 15°) + \mathcal{N}(270°, 15°)\big)$, where $\mathcal{N}$ denotes the normal distribution. Figure 4 illustrates the power output profiles of 24 houses in the PV-EW experiment over five days. This figure demonstrates that the curves have noticeable differences; however, they show similar

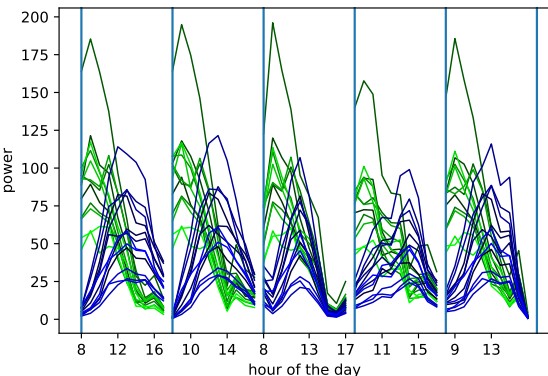

Figure 4: Power output profile of 24 houses in the PV-EW experiment over five days in June 2018, where each line represents the PV generation of one house. Green and blue curves correspond to houses facing the east and the west respectively. Although the curves have noticeable differences, there are consistent trends present in the data.

trends. As expected, the houses facing east or west are more similar to one another, while the differences among different sub-populations are higher. For instance, the peak production of the houses facing the east occurs in the morning, whereas the peak for those facing the west occurs in the afternoon.

To ensure that each client's dataset is identically distributed, despite the variations in meteorological patterns throughout the year, we filter the dataset down to the months of June and July. We employ two distinct training datasets for our analysis. The first dataset comprises the initial two weeks of June 2018, which provides a total of 150 samples for each client. The second dataset encompasses the data from both June and July 2018, resulting in 610 training samples per client. For all experiments, the test dataset consists of the data from June and July 2019. Furthermore, we exclude nighttime data points when the PV generation is zero. To normalize the data, we standardize each house's features and output, setting the mean to zero and the standard deviation to one.

**Features.** The dataset contains various meteorological features, including solar beam and diffuse irradiances, temperature, wind speed, solar altitude, time, and date. Solar beam and diffuse irradiances are particularly valuable in predicting PV generation, but their measurement can be costly. To reflect a realistic scenario, we assume that all households have access to the irradiance data recorded at a specific location in the city, such as a weather station. However, they are unaware of the specific irradiance values at their own houses, which may differ from the weather station. The temperature and wind speed may vary slightly among different houses in the dataset. Additionally, some houses may experience intermittent shadows caused by nearby trees or buildings, which can appear and disappear at certain times of the day. These shadows are considered as noise, and no recorded feature in the dataset provides explicit information about their occurrence or characteristics.

In addition to the meteorological features, we also incorporate autoregressors in our analysis. Autoregressors are advantageous in time-series prediction tasks and aim to capture the temporal dependencies in the data. To determine the autoregressors to include in our analysis, we utilize the partial auto-correlation function (PACF). The PACF measures the correlation between a time series and its lagged values while controlling for the correlations with all shorter lags. We select autoregressors with the highest values of the PACF among the lagged values up to two weeks ago. By focusing on these highly correlated autoregressors, we aim to capture the most relevant information from the past time steps.

### 8.5.2 SYNTHETIC DATASET

We examine a bimodal dataset where the data for each client is generated by sampling a function from one of two GP priors. Each GP prior is characterized by a polynomial mean function of order 7 and an SE kernel function. The two modes have distinct polynomial means and length scales associated

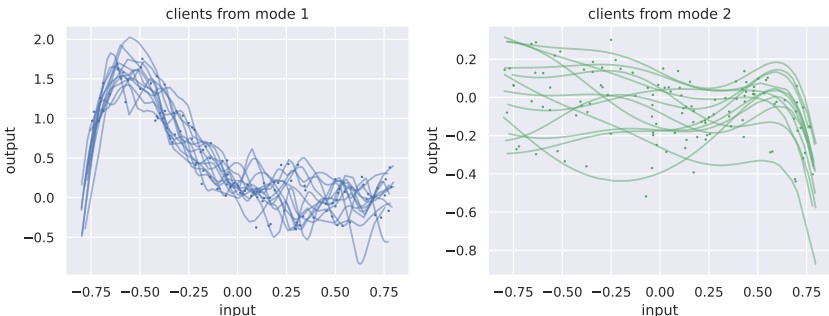

Figure 5: Synthetic dataset for a total of 24 clients, with 12 clients belonging to the first mode and another 12 clients belonging to the second mode. The solid lines represent the unknown true data-generating distribution for each client. Each client has a training dataset with 10 samples. The solid lines in the figure represent the true data-generating distribution, which is unknown to us. The points on the graph represent the noisy samples in each client's training dataset.

with the SE kernel. Additionally, Gaussian noise is introduced into the generated data to account for measurement errors and other sources of variability. Figure 5 illustrates the dataset for a total of 24 clients, with 12 clients belonging to the first mode and another 12 clients belonging to the second mode, where each client has 10 training samples.

## 8.6 EXPERIMENTS DETAILS

### 8.6.1 REGRESSION EXPERIMENTS DETAILS

**Baselines details.** In the Vanilla approach, the GP hyperparameters for each client are tuned by maximizing the LML of that specific client, without using FL. For pooled GP, we use inducing points to handle computational issues due to the large number of samples involved in this approach. pFedGP optimizes the average LML across clients to obtain the deep kernel hyper-parameters. We adapted pFedGP, originally designed specifically for classification, to suit our regression task. With this adaptation, pFedGP is a special case of our algorithm when using a single prior in SVGD and a very wide hyper-prior. We try pFedGP with a zero GP mean, as originally done in Achituve et al. (2021), and a NN mean and do not use inducing points. We apply pFedBayes (Zhang et al., 2022) only to the FEMNIST data as the authors only consider classification tasks in their experiments.

**Hyper-parameter tuning.** We perform hyper-parameter tuning for all baselines and our method using 5-fold cross-validation. For all neural networks, we explore structures with the same number of neurons per layer. The number of neurons per layer can take values of $2^n$ for $n \in 1, \cdots, 6$, and we consider 2 or 4 hidden layers. For PAC-PFL, we employ 4 SVGD particles and set $k = 4$. The parameter $\beta$ is set equal to the number of samples for each client, $\beta = m_i$. To determine the value of $\tau$, we conduct a search for values greater than $1/(1 + n\beta)$, ensuring that $\lambda > n_2 + v$ and thus, satisfying the assumption outlined in Theorem 4.1.

The employed hyper-prior is a multivariate Gaussian distribution with a diagonal covariance matrix. It is a distribution over the weights and biases of the mean and kernel neural networks, as well as the noise standard deviation of the likelihood. In all PV experiments, we set the hyper-prior mean for the neural network weights and biases to 0 and the hyper-prior mean for the noise standard deviation to 0.4. These choices help prevent overfitting.

**Results.** In this section, we evaluate our algorithm and compare it to the baseline methods introduced earlier. We assess the performance based on the RSMSE and CE error metrics for both existing and new clients. To obtain these metrics for a given method, we calculate the RSMSE and CE, where applicable, for each existing and new client using a fixed random seed. Then, we compute 4 values for the average RSMSE and CE metrics across the existing and new client groups. We performed the analysis five times using different random seeds and calculated the 95% confidence interval for the sample mean of RSMSE and CE over both the existing and new clients.

Table 2: RSMSE of existing clients for various methods. We report $95\%$ confidence interval for the average test RSMSE across all existing clients. Our proposed algorithm, PAC-PFL, demonstrates superior performance compared to all other baselines across all datasets.

| Method | PV-S (150) | PV-EW (150) | PV-S (610) | PV-EW (610) | Synthetic (10) |
|---|---|---|---|---|---|
| PAC-PFL | **0.429 ± 0.016** | **0.412 ± 0.008** | **0.425 ± 0.014** | **0.403 ± 0.008** | **0.582 ± 0.048** |
| pFedGP | 0.479 ± 0.038 | 0.452 ± 0.013 | 0.514 ± 0.034 | 0.457 ± 0.012 | 0.802 ± 0.147 |
| MTL | 0.445 ± 0.000 | 0.426 ± 0.001 | 0.441 ± 0.005 | 0.414 ± 0.000 | 0.851 ± 0.150 |
| MAML | 0.568 ± 0.023 | 0.523 ± 0.027 | 0.580 ± 0.016 | 0.524 ± 0.015 | 0.976 ± 0.129 |
| Vanilla | 0.683 ± 0.024 | 0.626 ± 0.033 | 0.488 ± 0.044 | 0.458 ± 0.015 | 0.727 ± 0.067 |
| Pooled | 0.484 ± 0.034 | 0.467 ± 0.045 | 0.492 ± 0.022 | 0.461 ± 0.030 | 2.328 ± 0.545 |

Table 3: RSMSE of new clients for various methods. We report $95\%$ confidence interval for the average test RSMSE across all new clients. Our proposed algorithm, PAC-PFL, outperforms other baselines in the majority of cases. The column name indicates the number of training samples specifically for the existing clients. However, the number of training samples for new clients remains constant at 150. In the Vanilla case, where each client trains their model individually, the size of the training data for the existing clients does not impact the performance for the new clients. Hence, the RSMSE values for the new clients employing the Vanilla method is the same in both the PV-S (150) and PV-S (610) datasets (similarly for PV-EW (150) and PV-EW (610)).

| Method | PV-S (150) | PV-EW (150) | PV-S (610) | PV-EW (610) | Synthetic (10) |
|---|---|---|---|---|---|
| PAC-PFL | **0.432 ± 0.018** | **0.417 ± 0.011** | **0.433 ± 0.012** | 0.430 ± 0.010 | **0.654 ± 0.024** |
| pFedGP | 0.464 ± 0.042 | 0.447 ± 0.017 | 0.513 ± 0.053 | 0.457 ± 0.012 | 0.874 ± 0.408 |
| MTL | 0.440 ± 0.001 | 0.423 ± 0.001 | 0.449 ± 0.002 | **0.430 ± 0.000** | 0.885 ± 0.123 |
| MAML | 0.546 ± 0.029 | 0.522 ± 0.013 | 0.552 ± 0.014 | 0.530 ± 0.021 | 1.107 ± 0.284 |
| Vanilla | 0.687 ± 0.020 | 0.632 ± 0.053 | 0.687 ± 0.020 | 0.632 ± 0.053 | 0.763 ± 0.046 |
| Pooled | 0.477 ± 0.029 | 0.469 ± 0.046 | 0.481 ± 0.022 | 0.460 ± 0.028 | 2.321 ± 0.371 |

The results are presented in Tables 2 to 5. In each column, the number after the dataset name represents the number of training samples per client. As shown in Tables 2 and 3, our method outperforms other approaches in the majority of cases in terms of RSMSE for both existing and new clients. Furthermore, we observe that our method's performance improves with an increase in the number of training samples per client.

Concerning CE, reported in Tables 4 and 5, PAC-PFL consistently exhibits low CE values across all scenarios, establishing itself as the top-performing or closely competitive method. This observation aligns with findings in Rothfuss et al. (2021), where the authors noted that CE improvement is notable when the meta-learning tasks exhibit greater similarity. In our own experiments, given the high client heterogeneity present in both PV-EW and PV-S, the improvements in CE are comparatively modest.

We believe there are three reasons why PAC-PFL outperforms other baselines. First, by learning the prior with FL and treating posterior inference as personalization, PAC-PFL exhibits a high level of adaptability to individual patterns. Second, PAC-PFL's ability to learn multiple priors enables it to effectively model clients with heterogeneous data distributions. Finally, the regularization of the hyper-posterior towards the hyper-prior in PAC-PFL helps prevent overfitting and allows for learning more complex prior means even with limited training data. For example, when selecting the GP prior mean structure through cross-validation, it is observed that the best GP mean for pFedGP is a linear function, while for PAC-PFL it is a 2-layer neural network with 32 neurons per layer. The use of a more complex prior mean in PAC-PFL allows for the potential capture of more intricate patterns in the data.

### 8.6.2 CLASSIFICATION EXPERIMENTS DETAILS

**Baselines details.** In our experiments, we utilize the original software by Achituve et al. (2021) for pFedGP and do not use inducing points. Since the source code for pFedBayes has not been publicly

Table 4: CE of existing clients for various methods. We report $95\%$ confidence interval for the average test CE across all existing clients.

| Method | PV-S (150) | PV-EW (150) | PV-S (610) | PV-EW (610) | Synthetic (10) |
|--------|-----------|-------------|-----------|-------------|----------------|
| PAC-PFL | $0.072 \pm 0.003$ | $\mathbf{0.041 \pm 0.002}$ | $0.068 \pm 0.002$ | $0.044 \pm 0.003$ | $0.140 \pm 0.017$ |
| pFedGP | $\mathbf{0.067 \pm 0.020}$ | $0.057 \pm 0.006$ | $0.066 \pm 0.002$ | $0.052 \pm 0.007$ | $\mathbf{0.119 \pm 0.012}$ |
| Vanilla | $0.122 \pm 0.008$ | $0.120 \pm 0.011$ | $\mathbf{0.036 \pm 0.002}$ | $\mathbf{0.028 \pm 0.001}$ | $0.178 \pm 0.011$ |
| Pooled | $0.256 \pm 0.004$ | $0.260 \pm 0.010$ | $0.263 \pm 0.005$ | $0.266 \pm 0.010$ | $0.326 \pm 0.036$ |

Table 5: CE of new clients for various methods. We report the $95\%$ confidence interval for the average test CE across all new clients. The CE values for the new clients employing the Vanilla method are not affected by the number of training samples for the existing clients, as indicated in the column names.

| Method | PV-S (150) | PV-EW (150) | PV-S (610) | PV-EW (610) | Synthetic (10) |
|--------|-----------|-------------|-----------|-------------|----------------|
| PAC-PFL | $0.073 \pm 0.005$ | $\mathbf{0.044 \pm 0.003}$ | $0.071 \pm 0.004$ | $\mathbf{0.044 \pm 0.002}$ | $0.165 \pm 0.008$ |
| pFedGP | $\mathbf{0.069 \pm 0.015}$ | $0.061 \pm 0.007$ | $\mathbf{0.064 \pm 0.007}$ | $0.056 \pm 0.008$ | $\mathbf{0.147 \pm 0.023}$ |
| Vanilla | $0.121 \pm 0.011$ | $0.120 \pm 0.010$ | $0.121 \pm 0.011$ | $0.120 \pm 0.010$ | $0.237 \pm 0.033$ |
| Pooled | $0.262 \pm 0.002$ | $0.263 \pm 0.010$ | $0.269 \pm 0.002$ | $0.266 \pm 0.014$ | $0.356 \pm 0.023$ |

released, we rely on an unofficial implementation[2]. Our implementations of FedAvg, MTL, and MAML are based on the code provided in Xie et al. (2023). For the Vanilla model, we follow the procedure described for the PV dataset. We utilize a Bayesian Convolutional Neural Network (BCNN) (Gal and Ghahramani, 2016) for all Bayesian approaches and a Convolutional Neural Network (CNN) for the frequentist methods.

**Hyper-parameter tuning.**    Hyper-parameter tuning for all baselines and our method is conducted using the cross-validation approach. In the case of PAC-PFL, we select $\beta$ and $\tau$ following the same procedure as employed for the PV dataset. We use 3 SVGD particles and a multivariate Gaussian hyper-prior with a diagonal covariance matrix.

**FEMNIST results.**    We evaluate PAC-PFL in two scenarios: first, with 40 clients, each having 20 training samples, and second, with the same 40 clients, each utilizing their entire available training dataset, averaging around 500 samples per client. In Table 6, we present the average accuracy and CE scores, together with $95\%$ confidence intervals, across five different random seeds for the existing clients. Remarkably, in the low-data scenario with only 20 samples per client, PAC-PFL consistently outperforms all other methods.

In the data-rich scenario, pFedGP demonstrates a slight advantage over PAC-PFL in terms of the mean, albeit with a considerably higher standard deviation, indicating sensitivity to initialization. Given the closely aligned means for both pFedGP and PAC-PFL and the fact that the confidence interval of PAC-PFL is encompassed within that of pFedGP, PAC-PFL is a more reliable method. It offers nearly identical mean performance. Additionally, pFedGP has a substantially higher computational cost than PACA-PFL. It is worth noting that the authors of Achituve et al. (2021) propose alternative variants of pFedGP that trade off accuracy for reduced computational cost. However, for our experiments, we employ the original algorithm.

---

[2]https://github.com/AllenBeau/pFedBayes

Table 6: Accuracy and calibration error on the FEMNIST classification dataset. We evaluate the metrics in two scenarios: first, with 40 clients, each having 20 training samples, and second, with the same 40 clients, each utilizing their entire available training dataset, averaging around 500 samples per client. The number after the metric name signifies the per-client training sample size. For each metric, the sample mean and 95% confidence interval using 5 random seeds is reported. While it is theoretically possible to calculate CE for pFedBayes, it is not included in the table as the available software only provides prediction means and lacks variances required for CE computation.

| Method | Accuracy (20) | CE (20) | Accuracy ($\approx 500$) | CE ($\approx 500$) |
|---|---|---|---|---|
| PAC-PFL | **0.942 ± 0.026** | **0.080 ± 0.013** | **0.971 ± 0.016** | 0.046 ± 0.007 |
| pFedGP | 0.836 ± 0.020 | 0.096 ± 0.039 | **0.977 ± 0.066** | **0.022 ± 0.008** |
| pFedBayes | 0.870 ± 0.020 | - | 0.883 ± 0.024 | - |
| FedAvg | 0.881 ± 0.017 | - | 0.967 ± 0.005 | - |
| MTL | 0.758 ± 0.037 | - | 0.875 ± 0.001 | - |
| MAML | 0.820 ± 0.067 | - | 0.881 ± 0.026 | - |
| Vanilla | 0.819 ± 0.013 | 0.120 ± 0.014 | 0.925 ± 0.018 | 0.325 ± 0.062 |
| Pooled | 0.894 ± 0.043 | 0.106 ± 0.048 | 0.943 ± 0.021 | 0.073 ± 0.033 |

