# OpenReview forum: "Personalized Federated Learning of Probabilistic Models: A PAC-Bayesian Approach"
_ICLR.cc/2024/Conference — Submitted to ICLR 2024_

### Official Review · Reviewer_5CVc · 2023-10-28

**Soundness:** 3 good
**Presentation:** 2 fair
**Contribution:** 2 fair
**Rating:** 3
**Confidence:** 4

**Summary:**

The paper studies the problem of federated learning from clients with highly heterogeneous data distribution and small datasets. To achieve this problem, the paper propose a PFL algorithm named PAC-PFL for learning probabilistic models within a PAC-Bayesian framework. The PAC-PFL learns a shared hyper-posterior in a federated manner, which clients use to sample their priors for personalized posterior inference. Both theoretical analysis and empirical results are provided to show the effectiveness of the proposed method.

**Strengths:**

1. The paper addresses a lot of issues for PFL, such as small datasets, highly heterogeneous data distribution, uncertainty calibration and new clients, which are critical problems.
2. The paper extensively provides both theoretical analysis and empirical results to show the effectiveness of the proposed method.

**Weaknesses:**

1. The studied problems are not well-driven and illustrated. For example, the descriptopm of uncertainty calibration is insufficient which is unfriendly for new readers. The studied issues, such as small datasets, highly heterogeneous data distribution, uncertainty calibration and new clients, should be further organized and summarized.
2. The novelty of the proposed method PAC-PFL is limited, since it seems that the PAC-PFL only combine some techniques, such as PAC-Bayesian, FedAvg and SVGD.
3. The related works[1-3] for tackling uncertainty calibration and FL for small datasets are omitted.
[1] Guo C, Pleiss G, Sun Y, et al. On calibration of modern neural networks. ICML, 2017: 1321-1330.
[2] Minderer M, Djolonga J, Romijnders R, et al. Revisiting the calibration of modern neural networks. NeurIPS, 2021, 34: 15682-15694.
[3] Fan C, Huang J. Federated few-shot learning with adversarial learning. WiOpt, 2021: 1-8.

**Questions:**

1. For experiments results, almost all FL methods usually outperform the Pooled GP baseline, which is strange and should be further explained.
2. In experiments results of tables and figures, the reported baselines are generally different with each other, which is confusing.
3. The studied problems are not well-driven and illustrated. For example, the description of uncertainty calibration is insufficient which is unfriendly for new readers. The studied issues, such as small datasets, highly heterogeneous data distribution, uncertainty calibration and new clients, should be further organized and summarized.
4. The novelty of the proposed method PAC-PFL is limited, since it seems that the PAC-PFL only combine some techniques, such as PAC-Bayesian, FedAvg and SVGD.

---

> ### Author Response · Authors · 2023-11-17
> **Response to Reviewer 5CVc - Part 1/2**
>
> Dear reviewer 5CVc,
>
> We appreciate your valuable feedback and thoughtful comments. Below, please find our answers to all your questions and concerns.
> We have submitted a revised version of the paper, encompassing the majority of the reviewers' comments. The main changes in the text appear in blue fonts. Should the paper be accepted, a comprehensive updated version will subsequently be submitted, addressing any outstanding comments. These updates, while relatively straightforward, were not included during the rebuttal phase due to timing constraints and are highlighted within specific responses.
>
> **1. The description of uncertainty calibration is insufficient which is unfriendly for new readers. The related works[1-2] for tackling uncertainty calibration are omitted.**
>
> We use calibration error as a metric for evaluating our trained model. Since the computation of this metric is not a central focus of our paper, we have restricted our discussion to citing relevant works (Rothfuss et al., 2021; Kuleshov et al., 2018), from which we derived the necessary formulas. The calibration formula employed for classification is consistent between (Kuleshov et al., 2018) and the first reference you provided, (Guo et al., 2017). However, it is worth noting that (Kuleshov et al., 2018) additionally present a metric for calibration error in regression, which prompted us to cite this work.
>
> We acknowledge that including a subsection in the appendix containing the formulas for computing the calibration error will enhance the completeness and self-sufficiency of our exposition. We will incorporate this in the final version and cite the references that you provided. We appreciate your feedback, and we hope that this enhancement will make our work more clear.
>
> **2. The studied issues, such as small datasets, highly heterogeneous data distribution, uncertainty calibration, and new clients, should be further organized and summarized.**
>
> We thank you for your comment and recognize that the addressed challenges might appear scattered over the introduction. We address four main challenges, which are: (c1) quantifying epistemic uncertainty, (c2) handling highly heterogeneous clients, (c3) considering clients possessing small datasets, and (c4) accounting for collecting new data over time. To enhance clarity, we are now enumerating these challenges in a paragraph in the introduction, assigning them labels (c1)-(c4). These labels are consistently used throughout the paper when referring to these challenges. Furthermore, in the experiments section, we now explicitly demonstrate how each experiment relates to these challenges. We invite you to refer to the updated version of the paper for further details.

---

> ### Author Response · Authors · 2023-11-17
> **Response to Reviewer 5CVc - Part 2/2**
>
> **3. The novelty of the proposed method PAC-PFL is limited since it seems that the PAC-PFL only combines some techniques, such as PAC-Bayesian, FedAvg, and SVGD.**
>
> While it is true that our method hinges on the existing pillars such as the PAC-Bayesian theory, FedAvg, and SVGD, there are substantial differences between our approach and each of these methods that we would like to highlight in the following.
> 1. Innovative formulation: We propose a rigorous framework for personalized learning by formulating a shared hyper-posterior and regarding posterior inference by each client as the personalization step (refer to Fig. 1 in the paper). Unlike other federated learning papers that employ PAC bounds solely for analyzing generalization, our approach uniquely integrates the PAC-Bayesian theory into algorithm design. Particularly, we specify the hyper-posterior by minimizing a PAC-Bayesian bound. We believe this formulation proposes an original and innovative framework.
> 2. Theoretical advances: Under this formulation, we establish a novel PAC-Bayesian bound at the server level in Theorem 4.1. This bound isn't a straightforward deduction from existing bounds, as reflected in our proof. The PAC-Bayesian theory has been classically developed for a single dataset of i.i.d. samples. However, our framework deals with multiple datasets from heterogeneous clients, necessitating the derivation of non-trivial extensions of the existing PAC-Bayesian bounds. These extensions, presented in our paper, require entirely new proofs and are absent in current literature.
> 3. Implementation: In our practical algorithm, we employ SVGD as a method to approximate the hyper-posterior. However, the critical aspect lies in determining the relevant probability distribution to approximate, which is the optimal hyper-posterior specified by our formulation and theoretical analysis. The structure of our algorithm for learning the shared hyper-posterior resembles FedAvg, where clients send the gradients of a cost function with respect to their local data and the server aggregates these gradients for updating the shared model. However, the crucial distinction lies in defining the cost to minimize and accordingly, determining the computation of gradients which is enabled through our formulation and theoretical analysis. Finally, FedAvg lacks the personalization aspect that is fundamental in PFL and is realized through posterior inference in our framework.
>
>
> In conclusion, the combination of these tools for achieving a PFL method, in our opinion, is far from being obvious.
>
>
> **4. The related work [3] for tackling FL for small datasets is omitted.**
>
> The authors of [3] profoundly explore FL in scenarios where clients' datasets are small. However, they concentrate on training a global shared model within their framework, without incorporating personalization. We highlight that personalized FL (PFL) is the main focus of our paper. Given the substantial body of work in PFL, our literature review has predominantly delved into personalized approaches.
>
> We, however, recognize the relevance of [3] to our work, particularly in considering clients with small datasets. In the final version of the paper, we will cite this paper in the introduction. Thank you for bringing this reference to our attention.
>
> **5. For experiment results, almost all FL methods usually outperform the Pooled GP baseline, which is strange and should be further explained.**
>
> Please refer to our answer 3.1 to reviewer D4J2.
>
> **6. In experiment results of tables and figures, the reported baselines are generally different from each other, which is confusing.**
>
> Thank you for this comment. We fully agree that baselines across different experiments should be consistent as much as possible. However, they cannot be exactly the same, as detailed in the sequel.  There are three primary differences between the baselines used in regression and classification. Firstly, pFedBayes is exclusively applied to the FEMNIST dataset, as explained in Section 8.6.1, because the authors of pFedBayes only consider classification tasks in their experiments. Second, we did not apply FedAvg to the PV dataset due to its anticipated suboptimal performance stemming from the dataset's high level of heterogeneity. Finally, following your comment and the comment from reviewer D4J2,  we realized that Pooled BNN was missing on the FEMNIST dataset and added it to Table 6 of the paper. Particularly, the accuracy of the Pooled BNN model is $89.4$% when each client has $20$ training samples and $94.3$% when each client uses all the available data. The respective calibration errors are $0.106$ and $0.073$.  We believe that this will make the baselines more homogeneous.
> In summary, with reference to the new version of the paper, pFedGP, MTL, MAML, Vanilla, and Pooled baselines are consistent across all experiments, while pFedBayes and FedAvg are exclusively employed for classification tasks.

---

> > ### Comment · Reviewer_5CVc · 2023-11-20
> >
> > Thank you for your clear response. Most of my concerns have been addressed and I have also read other comments and detailed responses. However,  I still have the following concerns:  novelty and computation complexity of the proposed method.

---

> > > ### Author Response · Authors · 2023-11-20
> > > **Response to Reviewer 5CVc - Round 2**
> > >
> > > Dear reviewer 5CVc,
> > >
> > > Thank you for your reply. We discussed the novelty of PAC-PFL in our previous response to your third question. Below, please find further explanations about the computational complexity.
> > >
> > > The computational complexity of PAC-PFL is discussed in Appendix 8.3.7 and in our response to the second question from reviewer D4J2. Particularly, the computational complexity of one iteration of PAC-PFL for client $i$ when using BNNs is $\mathcal{O}(k L m_i)$, where $k$ is the number of SVGD particles, $L$ is the number of samples from the prior to approximate the log marginal likelihood, and $m_i$ is the number of samples of client $i$. Here, $\mathcal{O}$ is the big-oh notation for computational complexity.
> > >
> > > Importantly, the computational cost of PAC-PFL scales linearly with the number of samples per client, making it applicable to large datasets. Moreover, previous studies have shown that SVGD achieves satisfactory performance with a relatively small number of particles (Liu and Wang, 2016). For instance, we utilize 4 SVGD particles for the PV dataset and 3 particles for the FEMNIST dataset in our experiments. Given that the cost is linear in both the number of samples per client and the number of  SVGD particles, $m_i$ and $k$, and requiring only a few particles, the overall complexity of PAC-PFL remains manageable. Our experiment on the FEMNIST dataset, with approximately 500 samples per client validates the scalability of PAC-PFL to realistically large datasets.

---

### Official Review · Reviewer_HvmF · 2023-10-29

**Soundness:** 3 good
**Presentation:** 3 good
**Contribution:** 3 good
**Rating:** 8
**Confidence:** 3

**Summary:**

This paper considers the personalized federated learning (PFL) through the lens of hierachical PAC-Bayes, similar to a previously studied PAC-Bayesian framework for meta-learning. a hyper-posterior is learned by a data-indpendent hyper-prior and the data from all clients, and a personalized posterior for each client is learned by a data-dependent prior sampled from the hyper-posterior and that client's data. To handle the data-dependence of the prior, an assumption on differential privacy is made and verified for optimal hyper-posterior, which is a Gibbs distribution. Based on this framework, a PAC-PFL algorithm is then proposed that updates the hyper-posterior is updaetd via SVGD.

**Strengths:**

Personalized federated learning is an important task, and as far as I can tell, the technical results are sound.
The experimental results show the proposed algorithm have better personalized performance.

**Weaknesses:**

There is no particular weakness of the paper.
The algorithm presented in the paper has too little details, especially how is the hyper-prior/posterior formulated (Gaussian distribution over the parameters?), make it less readable without knowledge of Rothfuss et al.
Though in applications, it is intractable to acheive the optimal Gibbs hyper-posterior, and thus lead to some concerns of the theory. I guess it is still possible to claim differential privacy for finite number of SVGD udpates?

**Questions:**

Plese see weakness.

---

> ### Author Response · Authors · 2023-11-17
> **Response to Reviewer HvmF**
>
> Dear reviewer HvmF,
>
> We appreciate your valuable feedback and thoughtful comments. Below, please find our answers to all your questions and concerns.
> We have submitted a revised version of the paper, encompassing the majority of the reviewers' comments. The main changes in the text appear in blue fonts. Should the paper be accepted, a comprehensive updated version will subsequently be submitted, addressing any outstanding comments. These updates, while relatively straightforward, were not included during the rebuttal phase due to timing constraints and are highlighted within specific responses.
>
> **1. How is the hyper-prior formulated?**
> We select a multi-variate Gaussian distribution with a diagonal covariance matrix as the hyper-prior, as indicated in Table 1 in Appendix 8.3.6. In the main paper, we refer to this table, mentioning that it contains selection guidelines in Section 5 after presenting the federated algorithm.
>
> To further clarify the hyper-prior selection in the main part of the paper, we have included this information regarding the hyper-prior selection in Section 5, specifically in the paragraph discussing the federated algorithm (please see the updated version of the paper). We appreciate your feedback.
>
> **2. Is it possible to claim differential privacy for a finite number of SVGD updates?**
>
> As detailed in Section 5 within the paragraph labeled "SVGD at the server level," since SVGD is a deterministic approach, it compromises the inherent privacy of the optimal hyper-posterior established in Lemma 4.3. However, we address this by suggesting a privacy-preserving variant of PAC-PFL in Appendix 8.4  for a finite number of SVGD updates. To enhance clarity and better emphasize our proposal of the privacy-preserving algorithm in the appendix,  we have revised the paragraph titled "SVGD at the server level". Please refer to the updated paper for further details. Thank you for bringing this to our attention and we hope the updates provide greater clarity.

---

> > ### Comment · Reviewer_HvmF · 2023-11-23
> >
> > Thank you for answering my questions.

---

### Official Review · Reviewer_QnoR · 2023-10-31

**Soundness:** 2 fair
**Presentation:** 2 fair
**Contribution:** 2 fair
**Rating:** 5
**Confidence:** 3

**Summary:**

Personalized Federated Learning (PFL) tailors a global model to individual clients' data, especially useful for diverse clients. To overcome challenges in PFL with limited client data, PAC-PFL is introduced. PAC-PFL employs a PAC-Bayesian framework and differential privacy, collaboratively learning a shared hyper-posterior while preventing overfitting through a generalization bound. Empirical tests on heterogeneous datasets confirm that PAC-PFL delivers accurate and well-calibrated predictions.

**Strengths:**

1. PAC-PFL introduces a systematic, non-heuristic regularization of the hyper-posterior, allowing for the training of complex models without falling into overfitting.
2. This  approach accommodates the accumulation of fresh data over time.
3. It can be interpreted through Jaynes' principle of maximum entropy
4. Experiments confirm PAC-PFL's accuracy in heterogeneous and bimodal client scenarios, along with its ability for efficient transfer learning from small datasets.

**Weaknesses:**

1. As for baslines, only 1 and the latest of them are proposed in 2022, methods that were proposed in 2023 should also be considered.
2. One dataset seems not enough for demonstrate the scalability and generalization of the proposed framework.
3. The theoritical analysis is pretty solid. However, the experiments are not convincing and strong enough in contrast.

**Questions:**

1. More state-of-the-arts methods should be included in the experiment.
2. More datasets should be performed on to illustrate the generalization of proposed framework.
3. The percentage of experiment should be increased compared with the theoritical analysis.

---

> ### Author Response · Authors · 2023-11-17
> **Response to Reviewer QnoR - Part 1/2**
>
> Dear reviewer QnoR,
>
> We appreciate your valuable feedback and thoughtful comments. Below, please find our answers to all your questions and concerns.
> We have submitted a revised version of the paper, encompassing the majority of the reviewers' comments. The main changes in the text appear in blue fonts. Should the paper be accepted, a comprehensive updated version will subsequently be submitted, addressing any outstanding comments. These updates, while relatively straightforward, were not included during the rebuttal phase due to timing constraints and are highlighted within specific responses.
>
> **1. As for baselines, only 1 and the latest of them are proposed in 2022, methods that were proposed in 2023 should also be considered.**
>
> In the submitted manuscript, we focused on comparing our method with personalized approaches for learning probabilistic models. It is conceivable that we might have overlooked some methods that were published while we were finalizing the paper. In conducting a literature review following your comment, we encountered (Ozkara et. al., 2023), which introduces a PFL approach with an interpretation within the empirical Bayes framework. However, to the best of our understanding, this algorithm does not offer uncertainty quantification and lacks a specific discussion on clients with small datasets. Upon acceptance, we intend to cite this paper in the introduction and highlight the differences from our work. If you are aware of any other methods, we would greatly appreciate it if you could inform us.
>
> *(Ozkara et. al., 2023) Kaan Ozkara; Antonious M Girgis; Deepesh Data; and Suhas Diggavi. A Generative Framework for Personalized Learning and Estimation: Theory, Algorithms, and Privacy. in International Conference on Learning Representations (ICLR). 2023.*
>
> **2. One dataset seems not enough to demonstrate the scalability and generalization of the proposed framework.**
>
> We analyze three datasets: 1) the PV dataset for regression, 2) a polynomial synthetic dataset for regression, and 3) the FEMNIST dataset for classification. For the PV data, we experiment with diverse levels of heterogeneity. In both PV and FEMNIST datasets, we manipulate the number of samples per client. Consequently, we evaluate the performance of PAC-PFL across seven variants of these datasets and present the findings in Tables 2-6. Section 6 briefly introduces the datasets employed, and Appendix 8.5 provides additional details. In the updated version of the paper, we have provided a list of all datasets at the beginning of the experiments section to ensure clarity regarding the datasets we utilize.
>
> With respect to scalability, our experiment on the FEMNIST dataset with around $500$ samples per client confirms that PAC-PFL scales reasonably well to realistically large datasets.
>
> To enhance the richness of our experiments, we have introduced a new experiment utilizing the EMNIST dataset and the Dirichlet partitioning technique for introducing heterogeneity. The accuracy results for our method and some of the baselines are as follows: PAC-PFL ($87.1$%), FedAvg ($82.6$%), pFedBayes ($79$%), and Vanilla ($71$%).  These results highlight the superior performance of PAC-PFL in handling the high heterogeneity introduced by the Dirichlet partitioning method. If the paper is accepted, we will dedicate a subsection to discussing the EMNIST dataset, accompanied by a comprehensive table showcasing the performance metrics of all baseline models. For detailed information about this experiment and its results, kindly refer to our response to question 3.2 from reviewer D4J2.
>
> By incorporating datasets of distinct natures, i.e., regression and classification, and by systematically varying the characteristics of each dataset, such as the heterogeneity level and the size of the datasets, we believe that our experiments effectively demonstrate the capabilities of our framework.

---

> ### Author Response · Authors · 2023-11-17
> **Response to Reviewer QnoR - Part 2/2**
>
> **3. The theoretical analysis is pretty solid. However, the experiments are not convincing and strong enough in contrast.**
>
> To ensure that our experiments effectively validate and showcase the proposed theory, we conducted simulations covering various aspects and features of the theoretical framework. To showcase the personalization capabilities, we utilized the highly heterogeneous and bi-modal PV-EW dataset. To highlight performance in scenarios with limited data, we varied the dataset size for both PV and FEMNIST datasets. To demonstrate the flexibility and generality of our method, we applied it to Gaussian Process regression and Bayesian Neural Network classification. By incorporating the new experiment on the EMNIST dataset, we substantiated the previously mentioned claims using a widely utilized benchmark.
>
> To improve clarity regarding the connection between theory and experiments, we are now labeling the considered theoretical challenges as (c1-c4) in the introduction. These labels are utilized to illustrate how each experiment aligns with these theoretical challenges. These updates have been implemented in the revised draft. We thank you for the feedback.
> We believe that the experiments offer substantial support for the theory, but If you observe any other methodological results that seem underrepresented in the experiments, please kindly bring them to our attention.

---

### Official Review · Reviewer_D4J2 · 2023-11-01

**Soundness:** 3 good
**Presentation:** 3 good
**Contribution:** 2 fair
**Rating:** 6
**Confidence:** 3

**Summary:**

This paper developed a PAC-Bayes framework for personalized federated learning. The PAC-PFL algorithm imposes a hyper-prior and a hyper-posterior on the server level. Based on a theoretical analysis (Theorem 4.1) similar to that of the client level (Theorem 3.1), the optimal hyper-posterior has a closed-form solution (Corollary 4.2.1). The final algorithm is based on several approximations (see Sec.5). Empirical studies on regression and classification problems show that PAC-PFL can outperform existing methods, especially when the sample size is small.

**Strengths:**

1. An algorithm developed from a theoretical perspective

2. Reasonable empirical results

3. Writing is mostly clear

**Weaknesses:**

1. The reason for using two samples per the client remains unclear. The algorithm requires two samples $S_i$ and $\tilde{S}_i$ as mentioned in the first paragraph of Sec.3. However, what they are used for specifically is not very clear. For example, in (8), shouldn’t the first $S_i$ be $S_i\cup\tilde{S}_i$ while the second one be $\tilde{S}_i$?

2. The computation complexity of the algorithm can be very high. For approximating the optimal hyper-posterior using a set of priors (see Sec.5), the communication overhead is increased from one to k, which can be unbearable for large models. It would be useful to see an ablation study on the choice of k.

3. Some experiment details require clarification

    3.1. we can see that PAC-PFL even outperforms the Pooled method in Tables 2 & 3, which is surprising as Pooled is essentially an oracle. Additional explanation would be helpful. Also, the Pooled method is missing for FMNIST.

    3.2. It is common to use Dirichlet partition (Marfoq et al., 2021, Wang et al., 2020) for other image datasets to simulate heterogeneous clients, so it would be more comparable to other baselines if the paper can include such an experiment.

Ref:
- Marfoq, O., Neglia, G., Bellet, A., Kameni, L. and Vidal, R., 2021. Federated multi-task learning under a mixture of distributions. Advances in Neural Information Processing Systems, 34, pp.15434-15447.
- Wang, H., Yurochkin, M., Sun, Y., Papailiopoulos, D. and Khazaeni, Y., 2019, September. Federated Learning with Matched Averaging. In *International Conference on Learning Representations*.

**Questions:**

Please clarify the questions mentioned in the Weaknesses section above.

---

> ### Author Response · Authors · 2023-11-17
> **Response to Reviewer D4J2 - Part 1/2**
>
> Dear reviewer D4J2,
>
> We appreciate your valuable feedback and thoughtful comments. Below, please find our answers to all your questions and concerns.
> We have submitted a revised version of the paper, encompassing the majority of the reviewers' comments. The main changes in the text appear in blue fonts. Should the paper be accepted, a comprehensive updated version will subsequently be submitted, addressing any outstanding comments. These updates, while relatively straightforward, were not included during the rebuttal phase due to timing constraints and are highlighted within specific responses.
>
> **1. The reason for using two samples per client remains unclear.**
>
> The rationale behind considering two distinct datasets for each client was explained in the fourth paragraph of the introduction and is detailed below. A selected client $i$ computes the global model’s update using locally available data denoted as $\mathcal{S}_{i}$. This is the first dataset considered for each client and aligns with the conventional definition of a client dataset in FL.
>
> In practical scenarios, it is desirable to conduct global model updates at a lower frequency while collecting data and executing model personalization more frequently. For example, in the Photovoltaic (PV) panels experiment, it is likely that households download the updated shared model from the server and transmit model updates back on a weekly basis while measuring new data and personalizing their models on a daily basis. Our approach accommodates different time scales for updating the shared model and for personalization.
>
> In between two consecutive updates of the global model, a client $i$ might collect a set of new data samples, denoted by $\tilde{\mathcal{S_{i}}}$. Although this data has not been utilized in training the shared model (i.e., the hyper-posterior), it can be leveraged by client $i$ for personalization. Considering an empty $\tilde{\mathcal{S_{i}}}$ set for all clients allows reverting to a typical FL framework. To the best of our knowledge, our work is the first to consider this second type of dataset, accounting for collecting new data over time. Another motivation for defining two datasets lies in providing a unified notation for existing clients and those joining the system later referred to as new clients. Given that a new client $\iota$ has never communicated with the server, $\mathcal{S_{\iota}}$ is an empty set. Conversely, $\tilde{\mathcal{S_{\iota}}}$ denotes the dataset that this new client has and can utilize for personalization.
>
> In summary, by defining two datasets for each client, we formulate an FL framework capable of accommodating data samples used for personalization but excluded from federated training. These excluded samples may be collected by existing clients after sending model updates to the server or by new clients who have not yet transmitted updates to the server. If the paper is accepted we will include a more detailed explanation for using two different datasets.
>
> **2. The computation complexity of the algorithm can be very high. The communication overhead is increased from one to $k$ which can be unbearable for large models. It would be useful to see an ablation study on the choice of $k$.**
>
> The computational complexity of PAC-PFL is discussed in Appendix 8.3.7. PAC-PFL leverages the closed-form solution for the optimal posterior, effectively reducing the computations performed by each client. Consequently, the primary computational load in each iteration lies in calculating or approximating the log marginal likelihood (LML) given each SVGD particle. Given that computing the LML is an inevitable step in most hyper-parameter selection methods for probabilistic models, the increase in the computational cost is due to considering multiple particles, growing linearly in the number of particles, $k$. Similarly, as you mentioned, the communication overhead also scales linearly with $k$.
>
> Notably, it has been demonstrated that SVGD achieves satisfactory performance with a relatively small number of particles (Liu and Wang, 2016). In our experiments, we employ 4 SVGD particles for the PV dataset and 3 particles for the FEMNIST dataset. If the paper is accepted, we will include a precise ablation study on the choice of the number of particles, $k$.
>
> Given that both the computational complexity and communication overhead scale linearly with the number of SVGD particles and considering the obtained satisfactory performance with a modest number of particles, the overall complexity remains manageable. Our experiment on the FEMNIST dataset with around $500$ samples per client confirms that PAC-PFL can scale to realistically large datasets. Further reducing the computational complexity is identified as a potential avenue for future research in Section 7.

---

> ### Author Response · Authors · 2023-11-17
> **Response to Reviewer D4J2 - Part 2/2**
>
> **3.1. PAC-PFL even outperforms the Pooled method in Tables 2 & 3, which is surprising as Pooled is essentially an oracle. Additional explanation would be helpful. Also, the Pooled method is missing for FMNIST.**
>
> The Pooled GP method, as introduced in Section 6, involves training a single shared model on a dataset that concatenates data from all clients in a data-centric manner. Due to the absence of personalization in this approach, its performance is expected to be low, especially for heterogeneous clients. The explanation has been included in Section 6 of the updated paper within the paragraph introducing the baselines (please see the updated version of the paper). We thank you for highlighting this issue.
>
> We acknowledge that the Pooled BNN results for the FEMNIST dataset were missing. The accuracy of the Pooled BNN model is $89.4$% when each client has $20$ training samples and $94.3$% when each client uses all the available data. The respective calibration errors are $0.106$ and $0.073$. These values have been added to Table 6 (please refer to the updated version of the paper). We thank you for bringing this to our attention.
>
>
> **3.2. It is common to use Dirichlet partition (Marfoq et al., 2021, Wang et al., 2020) for other image datasets to simulate heterogeneous clients.**
>
> Dirichlet partitioning is a method for artificially constructing a federated dataset with heterogeneous clients by partitioning a central and non-federated dataset. Each of these dataset partitions is then treated as the dataset for an individual client. This technique is useful when the central dataset lacks identifiers specifying the ownership of each sample by individual clients. For instance, the EMNIST dataset, a collection of hand-written characters, does not specify which writer produced each character.
>
> In contrast, the FEMNIST dataset provides information about the writer for each hand-written character, facilitating a natural partitioning of the data among clients. Consequently, there is no need for artificial data division to create client datasets, as one can directly use the data from each writer as an individual client's data. In this scenario, heterogeneity arises naturally from the inherent differences in handwriting styles among various writers. This explanation aligns with the experimental setup outlined in (Marfoq et al., 2021), where Dirichlet partitioning is employed for the EMNIST dataset, and natural partitioning by writers is utilized for the FEMNIST dataset.
>
> In response to your and reviewer QnoR's comments, we have performed a new experiment using the EMNIST dataset with Dirichlet partitioning. Following (Marfoq et al., 2021), we subsample $10$% of the EMNIST dataset and apply Dirichlet partitioning with the parameter $\alpha=0.4$ to synthesize heterogeneous clients. The objective of this experiment is to conduct a $64$-way classification task, recognizing hand-written characters that include both lower case and upper case English alphabet letters, as well as digits.
>
> The accuracy results for our method and some of the baselines are as follows: PAC-PFL ($87.1$%), FedAvg ($82.6$%), pFedBayes ($79$%), and Vanilla ($71$%). For comparison, (Marfoq et al., 2021) reported an accuracy of $83.5$% on the same dataset.  These results highlight the superior performance of PAC-PFL in handling the high heterogeneity introduced by the Dirichlet partitioning method. If the paper is accepted, we will dedicate a subsection to discussing the EMNIST dataset, accompanied by a comprehensive table showcasing the performance metrics of all baseline models.
>
>
> *(Liu and Wang, 2016) Qiang Liu and Dilin Wang. Stein variational gradient descent: A general purpose Bayesian inference algorithm. In Advances in Neural Information Processing Systems, volume 29. Curran Associates, Inc., 2016.*

---

> > ### Comment · Reviewer_D4J2 · 2023-11-21
> >
> > I thank the authors for the additional explanations. Some of my concerns have been addressed. However, for the first point, I understand these are different sets of data, but it remains unclear why two distinct sets are necessary and how they are used. For example, why does (7) use $\tilde{S}$ while (8) doesn't? Even in (7) alone, the expectation is only carried out for $\tilde{S}$ but not for $S$. Why is that?
> >
> > Overall, it seems that this split of datasets is an artifact of the algorithm instead of a real scenario encountered in practice.

---

> > > ### Author Response · Authors · 2023-11-22
> > > **Response to Reviewer D4J2 - Round 2**
> > >
> > > Dear reviewer D4J2,
> > >
> > > Thank you for your reply. Please find our answers to the points you have raised below.
> > >
> > > **1. Why two distinct sets are necessary and how they are used? It seems that this split of datasets is an artifact of the algorithm instead of a real scenario encountered in practice.**
> > >
> > > Let us present two examples illustrating practical scenarios where our setup is relevant, using the PV dataset.
> > >
> > > Consider a group of houses, each equipped with PV panels, participating in a PFL setup. Every house $i$ follows the setup described below:
> > > 1. Every week on day 1, house $i$ downloads the hyper-posterior from the server and sends back updates to it, calculated using dataset $\mathcal{S}_i$.
> > > 2. Throughout the week, house $i$ continuously measures its hourly PV generation, forming the dataset $\tilde{\mathcal{S}}_i$.
> > > 3. On a midweek day, such as day $5$, house $i$ personalizes its’ model using the data it had on the day $1$ and the data collected from day $1$ to day $5$, i.e. using the dataset $\mathcal{S}_i \cup \tilde{\mathcal{S}}_i$.
> > >
> > > In this example, employing two distinct datasets enables the integration of new data into the personalization process, enriching the personalized model. Alternatively, the house might choose to neglect the new data, $\tilde{\mathcal{S}}_i$, when personalizing its model. However, this approach is suboptimal as it disregards potentially valuable and fresh data.
> > >
> > > As a second example, consider a new house $\iota$ joining the PFL framework according to the following scenario:
> > > 1. It downloads the hyper-posterior trained on other houses' data from the server. The set $\mathcal{S}_\iota$ is empty as $\iota$ has never sent updates to the server.
> > > 2. It measures its hourly PV generation during 2 days, forming the dataset $\tilde{\mathcal{S}}_\iota$.
> > > 3. At the end of the second day, it personalizes its’ model using the dataset $\tilde{\mathcal{S}}_\iota$.
> > >
> > > In this example, considering two distinct datasets enables modeling a new house, which would not have been possible otherwise.
> > >
> > > The above examples highlight the benefits of defining two distinct datasets in practical scenarios. Notably, in applications where new data or new clients are irrelevant, our algorithm can still be utilized by considering an empty set $\tilde{\mathcal{S}}_i$ for all clients $i$.
> > >
> > > **2. Why (7) uses $\tilde{S}$? Why in (7), the expectation is only carried out for $\tilde{S}$ but not for $S$?**
> > >
> > > Equation (7) defines the server-level true risk which measures the average clients’ risk *after they personalize* a given hyper-posterior (see Fig. 1). As motivated by our response to question 1, personalization is carried out using both datasets $\mathcal{S}_i$ and $\tilde{\mathcal{S}}_i$. Because of using the set  $\tilde{\mathcal{S}}_i$ in personalization, it appears in Equation (7).
> > >
> > > The server computes an expectation w.r.t the sampling of new data in Equation (7) to account for the uncertainty associated with $\tilde{\mathcal{S}}_i$. The uncertainty arises from the fact that $\tilde{\mathcal{S}}_i$ denotes potential new samples that could be collected in the future and hence, are currently unknown. Conversely, there is no need for computing an expectation over the set $\mathcal{S}_i$, as it represents the current data held by clients and is not uncertain.
> > >
> > >
> > > **3. Why (8)  doesn't use $\tilde{S}$?**
> > >
> > > Equation (8) introduces the empirical counterpart of Equation (7) by replacing every unknown distribution $\mathcal{D}_i$ with a uniform distribution over $\mathcal{S}_i$. This substitution reduces the expectation taken over sampling new data from $\mathcal{D}_i$ in Equation (7) to uniformly sampling the new data from $\mathcal{S}_i$. Accordingly, the set $\mathcal{S}_i \cup \tilde{\mathcal{S}}_i$ in Equation (7) is replaced by $\mathcal{S}_i$ in Equation (8). Therefore, $\tilde{\mathcal{S}}_i$ does not exist in Equation (8). For a detailed proof of Equation (8) being the empirical counterpart of Equation (7), please refer to the proof of Theorem 4.1.
> > >
> > > We will add a more detailed explanation about points (2) and (3) to the paper if the paper is accepted.

---

### Comment · Area_Chair_y181 · 2023-11-10
**Authors-Reviewers discussion starts today, ends on Nov 22**

Dear authors and reviewers,

@Authors: please make sure you make the most of this phase, as you have the opportunity to clarify any misunderstanding from reviewers on your work. Please write rebuttals to reviews where appropriate, and the earlier the better as the current phase ends on Nov 22, so you might want to leave a few days to reviewers to acknowledge your rebuttal. After this date, you will no longer be able to engage with reviewers. I will lead a discussion with reviewers to reach a consensus decision and make a recommendation for your submission.

@Reviewers: please make sure you read other reviews, and the authors' rebuttals when they write one. Please update your reviews where appropriate, and explain so to authors if you decide to change your score (positively or negatively). Please do your best to engage with authors during this critical phase of the reviewing process.

This phase ends on November 22nd.

Your AC

---

### Meta-Review · Area_Chair_y181 · 2023-12-05

**Metareview:**

This meta-review is a reflection of the reviews, rebuttals, discussions with reviewers and/or authors, and calibration with my senior area chair. This paper explores a PAC-Bayes perspective of federated learning. Reviewers have stressed that the contributions appear too incremental compared to the existing literature, and that the supporting experiments are not conclusive enough to pass the acceptance bar at a competitive venue like ICLR. The discussion phase was somewhat frustrating as engagement from reviewers was not optimal, nevertheless I feel the manuscript requires a significant revision.

**Justification For Why Not Higher Score:**

The contribution feels incremental with respect to the existing literature and the experiments are not convincing enough to pass the ICLR bar.

**Justification For Why Not Lower Score:**

N/A

---

### Decision · Program_Chairs · 2024-01-16

Reject